**Measurement report: Chemical characteristics of PM$_{2.5}$ during typical biomass**
**burning season at an agricultural site of the North China Plain**
Linlin Liang[1], Guenter Engling[2,3], Chang Liu[1], Wanyun Xu[1], Xuyan Liu[4], Yuan Cheng[5], Zhenyu
Du[6], Gen Zhang[1], Junying Sun[1], Xiaoye Zhang[1]
[1] State Key Laboratory of Severe Weather & Key Laboratory for Atmospheric Chemistry, Chinese
Academy of Meteorological Sciences, Beijing 100081, China
[2] Division of Atmospheric Sciences, Desert Research Institute, Reno, NV 89512, USA
[3] Now at: California Air Resources Board, El Monte, CA 91731, USA
[4] National Satellite Meteorological Center, Beijing 100081, China
[5] School of Environment, Harbin Institute of Technology, Harbin 150001, China
[6] National Research Center for Environmental Analysis and Measurement, Beijing 100029 China
**Abstract:**
Biomass burning activities are ubiquitous in China, especially in North China, where there is an
enormous rural population and winter heating custom. Biomass burning tracers (i.e., levoglucosan,
mannosan and potassium (K$^+$)), as well as other chemical components were quantified at a rural site
(Gucheng, GC) in North China from 15 October to 30 November, during a transition heating season,
when the field burning of agricultural residues was becoming intense. The measured daily average
concentrations of levoglucosan, mannosan and K$^+$ in PM$_{2.5}$ during this study were $0.79 \pm 0.75$ μg
m$^{-3}$, $0.03 \pm 0.03$ μg m$^{-3}$ and $1.52 \pm 0.62$ μg m$^{-3}$, respectively. Carbonaceous components and biomass
burning tracers showed higher levels at nighttime than daytime, while secondary inorganic ions
were enhanced during daytime. An episode with high levels of biomass burning tracers was
encountered at the end of October, 2016, with high levoglucosan at 4.37 μg m$^{-3}$. Based on the
comparison of chemical components during different biomass burning pollution periods, it appeared
that biomass combustion can obviously elevate carbonaceous components levels, whereas no
essentially effect on secondary inorganic aerosols in the ambient air. Moreover, the
levoglucosan/mannosan ratios during different biomass burning pollution periods remained at high
values (in the range of 18.3 - 24.9), however, the levoglucosan/K$^+$ ratio was significantly elevated
during the intensive biomass burning pollution period (1.67) when air temperatures decreasing,
substantially higher than in other biomass burning periods (averaged at 0.47).
*Keywords*: Biomass burning; Organic tracers; Levoglucosan; Mannosan; Potassium

## 1. Introduction

Particulate air pollution is attracting more and more concerns in China because of their obvious
adverse impact on visibility reduction, as well as health implication and regional or global climate
change (Kanakidou et al., 2009; Pope and Dockery, 2006; Cheng et al., 2016). Carbonaceous species,
i.e., organic carbon (OC) and elemental carbon (EC), and water-soluble inorganic ions, e.g., $SO_4^{2-}$,
$NO_3^-$ and $NH_4^+$ are the major components of ambient aerosols (Liang et al., 2017; Du et al., 2014;
Zheng et al., 2015; Tan et al., 2016). Biomass burning (BB) emissions constitute a large source of
ambient particulate pollution, especially for carbonaceous components, i.e., primary organic carbon
(POC) and black carbon (BC) on global scale (Bond et al., 2004; Tang et al., 2018; Titos et al., 2017).
As an important aerosol component, black carbon from industrial and combustion emissions
contributes to the enhanced $PM_{2.5}$ (particulate matter with aerodynamic diameters less than 2.5 μm)
mass concentrations and influences regional radiative forcing (Chen et al., 2017). Fresh biomass
burning aerosol was found to be mainly comprised of carbonaceous species which typically
constitutes 50-60% of the total particle mass (Hallquist et al., 2009). Yao et al. (2016) identified
approximately half of carbonaceous aerosols being contributed by biomass burning at Yucheng, a
rural site in the North China Plain.
Biomass burning emissions also represent a potentially large source of secondary organic
aerosol (SOA). The precursors and formation pathways of SOA from biomass burning emissions
were investigated by extensive field observations (e.g., Zhu et al., 2015; 2017; Adler et al., 2011;
Zhang et al., 2010; 2015). Based on morphological particle analysis, Yao et al. (2016) investigated
the smoke emitted from biomass burning impacting SOA production. Sun et al. (2010) found that
phenolic compounds, which were emitted in large amounts from wood combustion, can form SOA
at high yields in aqueous-phase reactions. In addition, smoke from biomass burning can be
transported thousands of kilometers downwind from the source areas. Biomass burning aerosol from
Southeast Asia can be transported to China, Singapore and even further to North America (Liang et
al., 2017; 2020; Hertwig et al., 2015; Peltier et al., 2008). Based on molecular tracer measurements,

synoptic data as well as air mass back trajectory analysis, a fire episode was captured at a background site of East China with smoke advected from Southeast Asia (Liang et al., 2017).

The North China Plain (NCP) is one of the most polluted regions in China. Severe haze–fog of longer duration and more extensive coverage has occurred frequently in the NCP area, especially during the seasons of autumn and winter. NCP covers one quarter of China's cultivated land and yields 35% of the agricultural products in China (Boreddy et al., 2017). The rural population in NCP is also large and dense, and biomass burning activities are common in this region in form of cooking and heating. Intense fire activity typically occurs in October after the corn harvest. Abundant smoke is emitted from agricultural burning, i.e., residential biofuel combustion, open field burns, etc. Various field observations have investigated different aspects of biomass burning, e.g., seasonal variations, chemical and physical properties of smoke particles, spatial distribution, sources, transport, etc., in the NCP region (Cheng et al., 2013; Shen et al., 2018; Sun et al., 2013; 2016; Boreddy et al., 2017; Xu et al., 2019). However, these field investigations of the contribution of biomass burning to ambient aerosols in the NCP region were concentrated on the city of Beijing (Cheng et al., 2013; Zheng et al., 2015; Duan et al., 2004; Liang et al., 2016). Little field research about biomass burning was reported for rural areas in the NCP. In fact, biomass burning activities are common in the rural areas of the NCP region, and the resulting smoke aerosol can be transported to urban areas, e.g., the city of Beijing, resulting in haze episodic events.

In order to characterize the biomass burning pollution status within rural areas of the NCP region, multiple biomass burning tracers, i.e., levoglucosan, mannosan and $K^+$ in $PM_{2.5}$ sampled at a heavily polluted rural site in Hebei province were quantified during a typical biomass burning season, i.e., autumn-winter transition season, following the corn harvest. Combined with the analysis of other chemical components, it reveals different levels of biomass combustion pollution impacting on different types of chemical components in ambient aerosol. Meanwhile, based on the multi-analysis of biomass burning molecular tracers, back trajectory analysis, fire activity data and synoptic conditions, the results of this study demonstrate the biomass burning pollution status, as well as the formation process of severe biomass burning pollution episode in the rural atmosphere of North China. These results can provide valuable information about the biomass burning activities in all of Northern China.

## 2. Site description and experimental Methods

### 2.1 Site description and sampling

Samples were collected at a rural site, Gucheng (GC, 39°09'N, 115°44'E; 15.2 m a.s.l), located on a platform at the China Meteorological Administration farm in the town of Gucheng (GC site), approximately 110 km southwest of Beijing and 35 km north of the city of Baoding (population of about 5 million) in Hebei province, as shown in Figure S1. The station is surrounded by agricultural fields, with major crop species being corn and wheat. The dominant wind direction at GC is southwest and northeast during the study period. This site is upwind of Beijing, when the wind blows from the south or southwest, where heavily polluted cities and regions of Hebei province, i.e., Baoding, Shijiazhuang, Xingtai, Handan, are located. Thus, it is an appropriate station for representing the air pollution situation in the NCP region (Sheng et al., 2018; Chi et al., 2018; Xu et al., 2019; 2020; Kuang et al., 2020).

Daytime and nighttime $PM_{2.5}$ samples were collected from 15 October, 2016 to 23 November, 2016, by using $PM_{2.5}$ High-volume (Hi-Vol) sampler (GUV-15HBL1, Thermo Fisher Scientific CO., LTD), at the nominal flow rate of 1.13 $m^3$ $min^{-1}$. The daytime samples were collected from 07:00 to 19:00, while nighttime samples were collected from 19:00 to 07:00 local time of the next day. All $PM_{2.5}$ samples were collected on quartz fiber filters, prebaked at 850 °C for at least 5 h to remove carbonaceous material. A total of 33 couples of daytime/nighttime samples and 6 whole-day samples as well as 4 field blank samples were collected during the sampling period. The filters were stored at -20 °C after sample collection.

### 2.2 Experimental Methods

#### 2.2.1 Anhydrosugar and water-soluble inorganic ion analysis

The quartz filter samples were analyzed for biomass burning anhydrosugar tracers, i.e., levoglucosan and mannosan using an improved high-performance anion-exchange chromatography (HPAEC) method with pulsed amperometric detection (PAD) on a Dionex ICS-5000+ system. Levoglucosan and mannosan were separated by a Dionex Carbopac MA1 analytical column and guard column with an aqueous sodium hydroxide (NaOH, 480 mM) eluent at a flow rate of 0.4 mL $min^{-1}$. The detection limit of levoglucosan and mannosan was 0.002 mg $L^{-1}$ and 0.005 mg $L^{-1}$,

respectively. More details about the HPAEC-PAD method can be found elsewhere (Iinuma et al.,

115 2009).

The quartz filter samples were also analyzed for water-soluble inorganic ions by a Dionex ICS-
5000+ ion chromatograph, including $SO_4^{2-}$, $NO_3^-$, $NH_4^+$, $Cl^-$, $Ca^{2+}$, $Na^+$, $K^+$ and $Mg^{2+}$, and the
method detection limits for the individual ionic species were 0.18 µg $L^{-1}$, 0.15 µg $L^{-1}$, 0.03 µg $L^{-1}$,
0.048 µg $L^{-1}$, 0.08 µg $L^{-1}$, 0.01 µg $L^{-1}$, 0.01 µg $L^{-1}$, 0.008 µg $L^{-1}$, respectively. The cations were
separated on an Ionpac CS12 analytical column and CG12 guard column with a 20 mM
methanesulfouic acid as eluent at a flow rate of 1.0 mL $min^{-1}$, while the anions were separated on
an Ionpac AS11-HC column and AG11-HC guard column with 21.5 mM KOH eluent at a flow rate
of 1.0 mL $min^{-1}$. The water-soluble inorganic ion data were corrected by field blanks.
**2.2.2 Organic carbon/elemental carbon analysis**
OC and EC were measured on a punch (0.526 $cm^2$) of each quartz sample by a thermal/optical
carbon analyzer (DRI Model 2001, Desert Research Institute, USA), using the Interagency
Monitoring of Protected Visual Environments (IMPROVE) thermal evolution protocol with
reflectance charring correction. The analytical error of OC was within 10%, and one sample of every
10 samples was selected at random for duplicate analysis. The detection limit of OC was 0.82 µgC
$cm^{-2}$ (Liang et al., 2017).
**2.2.3 Gas online monitoring (i.e., NO, $NO_2$, $SO_2$, $O_3$, CO and $NH_3$)**
During this campaign, commercial instruments from Thermo Fisher Scientific Co., LTD were
used to measure $O_3$ (TE 49C), $NO/NO_2/NOx$ (Model 42CTL), CO (TE 48CTL), and $SO_2$
(TE43CTL), while $NH_3$ was measured by an ammonia analyzer (DLT-100, Los Gatos Research,
USA) at GC station. All measurement data quality was controlled according to standard gases (Xu
et al., 2019; Lin et al., 2011; Meng et al., 2018; Ge et al., 2018).
**2.2.4 Meteorological parameters**
The meteorological parameters, including air temperature, relative humidity (RH) and wind
speed at a 24-h resolution at the GC site are presented in Figure 1. During this campaign, the daily
average RH value was observed at $77 \pm 13\%$, with a range from 48% to 99%, while the daily wind
speed was observed with an average value of $1.07 \pm 1.14$ m $s^{-1}$, exhibiting moist and stable synoptic
conditions at this rural site during the autumn-winter transition season. Moreover, there was rare
precipitation during the sampling period at the GC site, except for two days, i.e., 20 and 27 October,
2016 (Figure 1).

**2.2.5 Back trajectory and fire spot analysis**

To characterize the transport pathways of the aerosol at the Gucheng site, back-trajectories
were calculated with the NOAA Hybrid Single-Particle Lagrangian Integrated Trajectory
(HYSPLIT) model via NOAA ARL READY Website (http://ready.arl.noaa.gov/HYSPLIT.php).
To investigate the influence of biomass burning activities in surrounding areas, fire hot spot
counts were obtained from the Fire Information for Resource Management System (FIRMS)
(available at https://firms.modaps.eosdis.nasa.gov/download/).

**2.2.6 Statistical analysis**

Statistical analysis of data, i.e., the correlation analysis between the concentrations of
levoglucosan, mannosan and $K^+$ at the Gucheng site during the sampling period were conducted
with the linear fitting method.

**3. Results and discussion**

**3.1 Characteristics of chemical components in PM$_{2.5}$**

In this study, the mass concentration of PM$_{2.5\text{-cal}}$ was reconstituted by the sum of carbonaceous
components ($1.6 \times$OC + EC) and inorganic ions ($SO_4^{2-}$ + $NH_4^+$ + $NO_3^-$ + $Cl^-$ + $Ca^{2+}$ + $Na^+$ + $K^+$ +
$Mg^{2+}$). Figure 1 describes the time-series variation obtained for daily PM$_{2.5\text{-cal}}$, OC, EC, biomass
burning tracers (levoglucosan, mannosan and $K^+$), ratios of levoglucosan/OC and meteorological
factors (temperature, RH, wind speed and planetary boundary layer (PBL) height) during the
sampling period. The average daily PM$_{2.5\text{-cal}}$ mass concentration in the autumn-winter transition
season at GC reached $137 \pm 72.4$ µg m$^{-3}$, ranging from 23.3 µg m$^{-3}$ to 319 µg m$^{-3}$ (Table 1, Figure
1a), which is higher than during the severe winter haze in January, 2013 at an urban site in Beijing
(121 µg m$^{-3}$) (Zheng et al., 2015). The mass concentrations of these chemical species during the day
are distributed as follows (from highest to lowest): OC > EC > $NO_3^-$ > $SO_4^{2-}$ > $NH_4^+$ > $Cl^-$ > $Ca^{2+}$ >
$K^+$ > $Na^+$ > $Mg^{2+}$. Organic matter (OM), calculated by multiplying OC values with a coefficient of
1.6, was the most abundant PM component, the daily average value of which was $70.4 \pm 49.6$ µg
m$^{-3}$, accounting for nearly half (46.7%) of PM$_{2.5\text{-cal}}$ mass, indicating obvious organic pollution at
the rural site in the North China Plain during the sampling season.
The measured daily average concentrations of biomass burning tracers, i.e., levoglucosan,
mannosan and $K^+$ in $PM_{2.5}$ during our study were $0.79 \pm 0.75$ μg m$^{-3}$, $0.03 \pm 0.03$ μg m$^{-3}$ and $1.52 \pm$
$0.62$ μg m$^{-3}$, respectively (Table 1). The ambient concentrations of levoglucosan in this study were
higher than those observed in the city of Beijing during the summer (averaged at $0.23 \pm 0.37$ μg m$^-$
$^3$, in the range of $0.06$ to $2.30$ μg m$^{-3}$) and winter (averaged at $0.59 \pm 0.42$ μg m$^{-3}$, in the range of
$0.06$ to $1.94$ μg m$^{-3}$) of 2010-2011 (Cheng et al., 2013). The biomass burning tracer levels and ratios
observed in this study and other field studies are summarized in Table S1. The highest
concentrations of levoglucosan in GC were observed on 31 October, 2016 with $4.37$ μg m$^{-3}$, which
is a sharp increase (over 30 times) of the minimum concentration ($0.14$ μg m$^{-3}$) during that period
(Figure 1c). Accordingly, the $PM_{2.5\text{-cal}}$ concentration during that period was also elevated (as high
as $236$ μg m$^{-3}$) (Figure 1a). Secondary inorganic aerosol (sulfate, $SO_4^{2-}$; nitrate, $NO_3^-$ and
ammonium, $NH_4^+$, SNA) species, were the major water soluble ions, accounting for 82.8% of total
water soluble ions, the daily average values of which were $10.5 \pm 6.87$ μg m$^{-3}$, $15.9 \pm 9.29$ μg
m$^{-3}$ and $10.9 \pm 5.51$ μg m$^{-3}$, respectively (Table 1). SNA species exhibited a synchronous temporal
trend (Figure 1c), while the $NO_3^-$ concentrations exceeded those of $SO_4^{2-}$ at the GC site, in contrast
to the results of previous studies, e.g., Tan et al. (2016), who found $SO_4^{2-}$ to be the dominant species
in $PM_{2.5}$ during winter in 2006 in Beijing. Similarly, Chi et al., (2018) also found
$NO_3^-$ concentrations exceeded those of $SO_4^{2-}$ at both Beijing and GC sites during the winter in 2016,
although they observed that $NH_4^+$ was the dominant component of SNA (the concentrations of
$SO_4^{2-}$, $NO_3^-$ and $NH_4^+$ were $14.0$ μg m$^{-3}$, $14.2$ μg m$^{-3}$, and $24.2$ μg m$^{-3}$, respectively).
**3.2 Day-night variations in the characteristics of $PM_{2.5}$ chemical components**
Carbonaceous components and biomass burning tracers exhibited higher levels during
nighttime than daytime, while secondary inorganic ions showed the opposite pattern, i.e., higher
concentrations during daytime than nighttime (Figure 2 and Figure S2). Besides, the gap of
carbonaceous components and anhydrosugars between daytime and nighttime (two-fold) was more
significant than for secondary inorganic ions. EC, POC are not subject to significant differences in
chemical reactions in ambient air between daytime and nighttime, and they will be mainly
influenced by the variations of the PBL height. In the night, the PBL height decreases, compressing

air pollutants into a shallow layer, and subsequently resulting in faster accumulation and higher concentrations of pollutants (Zheng et al., 2015; Zhong et al., 2018; 2019). The contributions of OM and EC to $PM_{2.5-cal}$ were observed to be higher at nighttime (53.9% and 16.6%) than daytime (43.8% and 13.7%) as well (Figure 3). Besides the influence from variations of the PBL height, the chemical degradation of levoglucosan may occur due to photochemical reaction in the ambient aerosols during daytime, further enlarging the gap of levoglucosan levels between daytime and nighttime (Sang et al., 2016; Gensch et al., 2018). Consequently, the contribution of levoglucosan to $PM_{2.5-cal}$ during daytime (0.45%) was observed to be considerably lower than that during nighttime (0.64%) (Figure 3). However, secondary inorganic ions have an important formation pathway, i.e., photochemical processing, during daytime. Thus, the secondary inorganic species ($SO_4^{2-}$, $NO_3^-$ and $NH_4^+$) were enhanced during daytime due to photochemical formation (Sun et al., 2013; Zheng et al., 2015; Wu et al., 2018). The mass contributions of $SO_4^{2-}$, $NO_3^-$ and $NH_4^+$ to $PM_{2.5-cal}$ were decreased from daytime (9.9%, 14.5% and 10.0%) to nighttime (6.5%, 9.6% and 7.1%) (Figure 3). Such an enhancement in secondary transformations during daytime is more evident in terms of the sulfur and nitrogen oxidation ratios (SOR and NOR, molar ratio of sulfate or nitrate to the sum of sulfate and $SO_2$ or nitrate and $NO_2$), which have been used previously as indicators of secondary transformations (Sun et al., 2013; Zheng et al., 2015). Both SOR and NOR during daytime were higher than those during nighttime (Figure S3), further confirming the elevated secondary formations of sulfate and nitrate during daytime.

In addition, the concentrations of other water-soluble inorganic ions, i.e., $K^+$ and $Cl^-$ during nighttime ($1.78 \pm 0.95$ μg m$^{-3}$ and $6.08 \pm 4.00$ μg m$^{-3}$) were higher than those in daytime ($1.43 \pm 0.54$ μg m$^{-3}$ and $4.33 \pm 2.30$ μg m$^{-3}$), while their contributions to $PM_{2.5-cal}$ were reversed, due to the significant accumulation and higher concentrations of pollutants during nighttime. As $Ca^{2+}$, $Mg^{2+}$ and $Na^+$, mainly emitted from primary natural sources, such as dust, soil resuspension and sea salt, are subject to more activity during the daytime and also influenced by the airflow dynamics, the contribution of those species in nighttime were lower than those during daytime, especially for $Ca^{2+}$, decreasing from 2.2% in daytime to 0.9% at nighttime (Figure 3).

### 3.3 Biomass burning episodes and the impacts on chemical PM2.5 characteristics

An episode with high biomass burning tracer levels was encountered on 31 October, 2016.

The concentrations of levoglucosan in $PM_{2.5}$ during this one-day episode (4.37 µg m$^{-3}$) were
significantly higher than those during typical transition season at the GC site (0.69 ± 0.47 µg m$^{-3}$)
(Figure 1d). Meanwhile, there was significant change in the meteorological conditions, i.e., the wind
direction changed from southwesterly to northerly winds (Figure S4). Northerly winds advected
cold and dry air masses, with the lowest hourly temperature observed at -5.3 °C (Figure S5). This
notable temperature decline before the commencing of the operation of the central heating systems
should have caused intense combustion activities for heating purposes at the rural site. Moreover,
the synoptic situation on 31 October, 2016 was under weaker turbulence with low PBL height and
small wind speeds (Figure 1f). These worsened meteorological conditions would further enhance
aerosol accumulation.
Here, we mainly distinguish four sub-periods based on daily levoglucosan concentrations
during the time frame from 15 October to 23 November, 2016. The four periods were separated as
follows: 15-30 October (Period I: Minor biomass burning), 31 October (Period II: Intensive biomass
burning), 1-14 November (Period III: Major biomass burning), 15-23 November (Period IV:
Heating season). Table 2 compares the concentrations of $PM_{2.5-cal}$ mass, chemical components and
gases at the GC site during these four periods, as well as the ratios between the intensive, major BB
periods and heating season to minor BB period. The level of levoglucosan during the intensive BB
episode II was about 12 times of that during the minor BB period I. $K^+$ and $Cl^-$, the common biomass
burning tracers utilized in many studies (Duan et al., 2004; Cheng et al., 2013), were also observed
with increased abundance during intensive BB episode II. When entering into November, the
weather was becoming cold, and thus combustion activities for heating in the rural areas commenced,
resulting in the ambient levels of levoglucosan to increase to 0.92 ± 0.47 µg m$^{-3}$ during period III,
about 3 times of those in Period I. The central heating systems in North China cities were operated
during period IV, and the ambient level of levoglucosan was observed at 0.96 ± 0.63 µg m$^{-3}$, which
was similar to that observed in period III.
The concentrations of OC and EC were also observed to be strongly elevated in period II (Table
2), and especially OC levels increased to 96.3 µg m$^{-3}$ during the intensive BB episode II, nearly 6
times of those during the minor BB period (16.2 ± 7.52 µg m$^{-3}$). The levoglucosan/OC ratio was
utilized to estimate the effect of biomass burning to ambient organic aerosols. Accordingly,
levoglucosan/OC ratios sharply increased to 0.045 during period II, which was noticeably higher
than during other periods in this study (Figure 1e). Moreover, this level is also higher than most of
the published field observations, i.e., at urban sites (Zhang et al., 2008; Cheng et al., 2013; Zhang
et al., 2014), rural sites (Sang et al., 2013; Ho et al., 2014; Pietrogrande et al., 2015; Mkoma et al.,
2013) and agricultural sites (Ho et al., 2014; Jung et al., 2014), yet lower than at an urban site in
northern Italy during winter time (in the range of 0.01 to 0.13) (Pietrogrande et al., 2015). This
illustrates that biomass combustion played an important role in organic aerosol pollution during the
intensive BB episode II. However, due to other emissions of OC enhanced during the major BB
episode (period III) and heating season (period IV), i.e., combustion of coal and biofuel for heating,
OC increased to a higher level ($55.2 \pm 17.1 \ \mu gC \ m^{-3}$ and $69.4 \pm 24.6 \ \mu gC \ m^{-3}$, respectively). Due to
the abundance of organic aerosols, the contribution from biomass burning emission was thereby
reduced and the levoglucosan/OC ratios during periods III and IV decreased to $0.016 \pm 0.005$ and
$0.014 \pm 0.006$, respectively, even lower than those observed in the minor BB period I ($0.025 \pm 0.008$).

271        Compared to the carbonaceous components, the concentrations of secondary inorganic aerosol

species ($SO_4^{2-}$, $NO_3^-$, $NH_4^+$) exhibited a different pattern, i.e., showing no obvious differences
between minor BB period I and other three periods. The ratios of $SO_4^{2-}$, $NO_3^-$, $NH_4^+$ during periods
II, III and IV to period I were all around 1.0 (Table 2), with no increasing trend. Moreover, the
relationships between levoglucosan and OC (and EC) were better than those between levoglucosan
and SNA during daytime and nighttime (Figure S3). The precursor gases of SNA, i.e., $SO_2$, NO,
$NO_2$ and $NH_3$, were observed to have an increasing trend when biomass burning was prevalent
during periods III and IV, with the ratios to period I arranged from 1.13 to 1.90 (Table 2). The time-
series variations of the gases ($SO_2$, $NO_x$, $NH_3$, CO and $O_3$) and PBL during the sampling period
are shown in Figure S4. The primary emission gases were exhibited negative relationships with PBL,
while $O_3$ exhibited obvious positive relationship with PBL (Figure S5). Combustion from different
fossil fuels (coal, gasoline, diesel, etc.) and biomasses (straws, woods, leaves, etc.) can all emit CO
into the atmosphere (Streets et al., 2003; Chantara et al., 2019; Merico et al., 2020). Due to the more
abundant combustion in the colder weather, the concentrations of CO also increased to $1.65 \pm 0.53$
ppm and $1.18 \pm 0.83$ ppm during the major biomass burning period III and the heating season period
IV, respectively.
The combustion of biomass, especially of agricultural residues (e.g., wheat and corn straw) is
very common in the rural areas in North China during the autumn-winter transition period. During
the autumn harvest season in North China, wheat and corn straw burning is common practice,
resulting in more abundant fire spots when entering into November than period I (Figure 4). The
intense biomass burning event on 31 October, 2016 was also supported by air mass back trajectory
analysis (Figure 5), performed with the TrajStat software. Based on the 48 h back trajectories at the
GC site at 00:00 (UTC time) on 1 November, 2016, the air mass at the GC site was restricted in the
region of Bejing-Tianjing-Hebei, the polluted area where fire spots were numerous. However, on
the previous and following day of this episode, i.e., 31 October and 2 November onward, the air
masses arriving at GC were advected from the northwest of Mongolia, where mostly desert areas
are present, with less farm land and rare biomass burning activities (Figure 5).
Mean percentiles of major components in $PM_{2.5}$ with respect to different BB pollution periods
at GC site during the sampling time are shown in Figure 6. With the variation of BB pollution
periods, the EC fraction seems to exhibit no obvious change during periods I, II and III, but slightly
increased during the heating season (period IV), while the OC fraction increased significantly from
34.0% during the minor BB period I elevated to 65.4% during the intense BB period II. The
contributions of sulfate, nitrate and ammonium to $PM_{2.5-cal}$ all decreased sharply from the minor BB
period to the intense period (Figure 6). This suggests that organic aerosol species become more
important during BB pollution periods, concerning their contribution to the $PM_{2.5-cal}$, while EC has
no such character. The OM percentage during intense BB period II was 65.4%, about double of that
during the minor biomass burning period (34.0%), indicating that there was a large fraction of OM
in $PM_{2.5-cal}$ originating from BB at the GC site during intensive BB period II. Opposite to OM,
contributions of secondary inorganic ions to $PM_{2.5-cal}$ significantly decreased with the BB pollution
becoming more severe. The contributions of $SO_4^{2-}$, $NO_3^-$ and $NH_4^+$ to $PM_{2.5-cal}$ during the minor
BB episode (11.6%, 20.5% and 12.5%) obviously declined during the intense BB episode (1.93%,
7.67% and 4.24%).

**3.4 Relationships among tracers during different biomass burning pollution periods**

In addition to pollution level information of biomass burning molecular tracers, the ratios between them could also be used to identify the different biomass types or indicate the burning formation processes of atmospheric aerosols. Levoglucosan and mannosan showed a good relationship during the entire sampling period (Figure 7a, $r = 0.97$, $p < 0.01$). The levoglucosan/mannosan ratios during minor, intense, major biomass pollution and heating season periods were observed at high values, i.e., 24.9, 24.1, 24.8 and 18.3 respectively (Table 2, Figure 7). Compared to the former three episodes (24.1 to 24.9, averaged at 24.6), the levoglucosan/mannosan ration during the heating season period (18.3) decreased by 25.6%. Based on source emission studies, the levoglucosan/mannosan ratios from crop residue burning, i.e., rice straw, wheat straw and corn straw, are similar and are characterized by high values (averaged at 29, in the range of 12 to 55) (Zhang et al., 2007; Engling et al., 2009; Cheng et al., 2013; Jung et al., 2014), yet overlapping with those from hardwood (averaged at 28, in the range of 11 to 146) (Bari et al., 2009; Jung et al., 2014) and grass burning ($18.2 \pm 10.2$) (Sullivan et al., 2008), while softwood is characterized by relatively lower levoglucosan/mannosan ratios (averaged at 4.3, in the range of 2.5 to 4.7) (Engling et al., 2006; Cheng et al., 2013; Jung et al., 2014). Subsequently, this declining trend in the levoglucosan/mannosan ratios during the heating season period was partly caused by the higher proportion of softwood combustion, which is characterized by relatively lower levoglucosan/mannosan ratios. According to the local habits, soft woods, e.g. China fir and pine are also commonly used as biofuels for stove heating in North China, since they allow sustained heating duration.

The concentrations of levoglucosan and $K^+$ during minor, major BB episode and heating season were correlated well (Figure 7b, $r = 0.84$, $p < 0.01$), while the red dot of period II being off from the fitted regression line. The levoglucosan/$K^+$ ratios during periods III and IV (0.51 and 0.53) were similar to those during a BB episode at an urban site in Beijing during winter time (levoglucosan/$K^+$ = 0.51) (Cheng et al., 2013). However, the levoglucosan/$K^+$ ratio during the intense BB period II increased to 1.67, which was significantly higher than that in typical straw combustion ($< 1.0$). Correspondingly, there was a significant drop in temperatures at the GC site

during period II, with the average daily temperature sharply decreasing from 7.5 °C on 30 Oct to
0.31 °C on 31 October, 2016, and the average temperature at night of 31 October even decreased to
-3.4 °C (Figure 1g). Hence, the combustion activities were apparently intense around the sampling
site for heating purposes. Compared to $K^+$, there is a large enrichment of levoglucosan in wood
burning emissions, based on the results from previous biomass source combustion studies (Engling
et al., 2006; Chantara et al., 2019). The influence of softwood and/or other materials from softwood,
which are commonly used as biofuels for stove heating in North China (Cheng et al., 2013; Zhou et
al., 2017), should be larger during this low temperature period. Moreover, levoglucosan/$K^+$ ratios
also can be influenced by combustion conditions, i.e., smoldering versus flaming burns. Biofuels
are typically subject to smoldering combustion condition in residential stoves for heating purposes
in the rural areas in North China, which was reflected in relatively higher levoglucosan/$K^+$ ratios
than during flaming combustion (Schkolnik et al., 2005; Lee et al., 2010).
**4. Summary and conclusion**
Anhydrosugars, including levoglucosan and mannosan, and water-soluble potassium ion were
employed as molecular tracers to investigate the characteristics of biomass burning activities as well
as chemical properties of ambient aerosols under different biomass burning pollution levels. The
measured daily average concentrations of levoglucosan, mannosan and $K^+$ in $PM_{2.5}$ during a typical
biomass burning season from 15 October to 30 November, 2016 were 0.79 ± 0.75 μg m$^{-3}$, 0.03 ±
0.03 μg m$^{-3}$ and 1.52 ± 0.62 μg m$^{-3}$, respectively. The concentrations of carbonaceous components
and biomass burning tracers were observed higher at nighttime than daytime, while the patterns of
secondary inorganic ions ($SO_4^{2-}$, $NO_3^-$ and $NH_4^+$) were opposite, since they were enhanced by
photochemical formation during daytime. An episode with extreme biomass burning tracer levels
was encountered on 31 October, 2016, with concentrations of levoglucosan as high as 4.37 μg m$^{-3}$.
Comparing the chemical compositions between different biomass burning periods, it was apparent
that biomass burning can considerably elevate the levels of organic components, while not showing
a significant effect on the production of secondary inorganic ions. Compared to the other biomass
burning episodes, the levoglucosan/mannosan ratios during the heating season period slightly
decreased, while levoglucosan/$K^+$ ratio during the intensive BB period was unusually higher than
those in the other three biomass burning periods.

*Data availability*. The data used in this study can be obtained from this open link: https://pan.baidu.com/s/11bKUZff1KJbzNVxS3VsLaA code: jvqx. It is also available from the corresponding author upon request (lianglinlin@cma.gov.cn).

*Author contributions.* LL designed conducted all observations and drafted the paper. GE revised the paper and improved the English writing. XL drew the Figure 4 and Figure 5. CL, WX, YC, ZD, GZ, JS and XZ interpreted the data and discussed the results. All authors approved the final version for publication.

*Competing interests.* The authors declare that they have no conflict of interest.

*Special issue statement.* This article is part of the special issue "In-depth study of air pollution sources and processes within Beijing and its surrounding region (APHH-Beijing) (ACP/AMT interjournal SI)". It is not associated with a conference.

*Acknowledgements.* This research is supported by the Beijing Natural Science Foundation (8192055) and CAMS Fundamental Research Funds (No. 2017Z011). The authors would like to acknowledge Yingli Yu and Ye Kuang for their help with $PM_{2.5}$ samples collection; Hongbing Cheng for help with chemical analyses.

*Financial support*. This research has been supported by the Beijing Natural Science Foundation (No. 8192055), State Environmental Protection Key Laboratory of Sources and Control of Air Pollution Complex (No. SCAPC201701) and Chinese Academy of Meteorological Sciences Fundamental Research Funds (No. 2017Z011).

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

Contributions and source identification of biogenic and anthropogenic hydrocarbons to
secondary organic aerosols at Mt. Tai in 2014, Environ. Pollut., 220, 863-
872, https://doi.org/10.1016/j.envpol.2016.10.070, 2017.

















**Table 1.** Average concentrations and the range of $PM_{2.5-ca1}$ and its chemical components, biomass burning
tracers (µg m$^{-3}$), gaseous species, ratios of OC/EC and levoglucosan /OC, as well as meteorological data
observed at GC site at daytime, nighttime and whole day, respectively, during the sampling period from
15 Oct to 23 Nov 2016.

| Species | Daytime (N = 34) | | Nighttime (N = 33) | | Whole period (N = 37)* | |
|---|---|---|---|---|---|---|
| | Average concentration | Range | Average concentration | Range | Average concentration | Range |
| $PM_{2.5-cal}$ | 117 ± 58.8 | 19.0 - 225 | 170 ± 116 | 21.1 - 465 | 137 ± 72.4 | 23.3 - 319 |
| OC | 26.8 ± 15.7 | 3.78 - 64.8 | 61.6 ± 49.5 | 2.88 - 175 | 44.0 ± 31.0 | 4.13 - 117 |
| EC | 13.4 ± 8.49 | 1.44 - 34.0 | 30.9 ± 28.5 | 2.21 - 129 | 21.7 ± 15.8 | 2.46 - 74.9 |
| TC | 49.3 ± 27.6 | 5.76 - 124 | 92.5 ± 73.6 | 5.10 - 289 | 65.8 ± 44.1 | 7.36 - 192 |
| OC/EC | 2.02 ± 1.26 | 1.09 - 3.31 | 2.25 ± 1.04 | 1.04 - 6.72 | 1.95 ± 0.60 | 0.83 - 3.10 |
| $SO_4^{2-}$ | 12.1 ± 9.31 | 1.65 - 39.7 | 9.02 ± 6.22 | 1.55 - 23.2 | 10.5 ± 6.87 | 1.66 - 29.5 |
| $NO_3^-$ | 16.9 ± 9.96 | 1.85 - 41.2 | 13.1 ± 8.52 | 1.56 - 38.0 | 15.9 ± 9.29 | 2.40 - 45.2 |
| $Cl^-$ | 4.33 ± 2.30 | 0.82 - 9.46 | 6.08 ± 4.00 | 0.62 – 16.0 | 4.90 ± 2.46 | 0.93 - 9.37 |
| $NH_4^+$ | 11.7 ± 6.76 | 1.84 - 26.0 | 10.0 ± 5.75 | 1.33 - 22.2 | 10.9 ± 5.51 | 1.99 - 25.4 |
| $K^+$ | 1.43 ± 0.54 | 0.20 - 2.64 | 1.78 ± 0.95 | 0.22 - 4.19 | 1.52 ± 0.62 | 0.50 - 2.96 |
| $Mg^{2+}$ | 0.26 ± 0.14 | 0.07-0.64 | 0.19 ± 0.09 | 0.06 - 0.38 | 0.14 ± 0.12 | 0.04 - 0.43 |
| $Ca^{2+}$ | 2.24 ± 1.01 | 1.02-4.75 | 1.56 ± 0.08 | 0.77 - 3.56 | 1.54 ± 0.90 | 0.49 - 3.84 |
| $Na^+$ | 0.44 ± 0.17 | 0.10 - 0.79 | 0.43 ± 0.24 | 0.10 - 1.31 | 0.42 ± 0.17 | 0.11 - 0.88 |
| $NO_3^-$ / $SO_4^{2-}$ | 1.67 ± 0.82 | 0.75 - 5.52 | 1.54 ± 0.57 | 0.74 - 3.50 | 1.65 ± 0.62 | 0.78 ± 3.96 |
| Levoglucosan | 0.57 ± 0.62 | 0.05 - 3.74 | 1.10 ± 0.99 | 0.05 - 4.82 | 0.79 ± 0.75 | 0.14 - 4.37 |
| Mannosan | 0.024 ± 0.023 | 0.00 - 0.14 | 0.05 ± 0.04 | 0.00 - 0.21 | 0.03 ± 0.03 | 0.00 - 0.18 |
| levoglucosan/OC | 0.018 ± 0.011 | 0.005 - 0.067 | 0.020 ± 0.010 | 0.004 - 0.047 | 0.020 ± 0.009 | 0.006 - 0.045 |
| NO (ppb) | 23.0 ± 14.7 | 2.07 - 56.0 | 45.9 ± 29.5 | 1.59 - 96.9 | 31.8 ± 18.3 | 1.81 - 68.5 |
| $NO_2$ (ppb) | 25.8 ± 10.4 | 8.18 - 51.6 | 29.3 ± 9.37 | 8.81 - 51.1 | 26.6 ± 8.74 | 8.62 - 51.4 |
| $SO_2$ (ppb) | 9.78 ± 4.96 | 3.11 - 22.5 | 9.63 ± 5.67 | 2.91 - 28.7 | 8.61 ± 4.04 | 3.37 - 20.4 |
| CO (ppm) | 0.96 ± 0.73 | 0.03 - 2.49 | 1.29 ± 1.04 | 0.02 - 3.26 | 1.05 ± 0.76 | 0.12 - 2.48 |
| $O_3$ (ppb) | 13.0 ± 9.10 | 1.42 - 41.84 | 5.00 ± 5.73 | 1.60 - 24.30 | 9.25 ± 5.78 | 1.67 - 24.0 |
| $NH_3$ (ppb) | 16.4 ± 11.3 | 1.68 - 46.2 | 18.3 ± 10.7 | 1.03 - 42.7 | 17.1 ± 9.88 | 1.46 - 44.4 |
| Temperature (°C) | 7.71 ± 4.01 | - 2.07-15.9 | 3.30 ± 4.69 | - 6.60 - 14.5 | 6.95 ± 4.58 | - 4.33 - 15.4 |
| Relative Humidity (%) | 68 ± 17 | 31 - 98 | 85 ± 14 | 34 - 100 | 77 ± 13 | 48 - 99 |
| Wind speed (m s$^{-1}$) | 1.43 ± 1.17 | 0.09 - 5.65 | 0.79 ± 1.55 | 0.03 - 7.19 | 1.07 ± 1.14 | 0.04 - 5.02 |

* Six whole-day samples were included in the data analysis of the "Whole period".






**Table 2.** Concentrations of chemical components in $PM_{2.5}$ aerosols as well as their ratios and gaseous
species collected at the GC site, during the four biomass burning periods (i.e., Minor, Intensive, Major
and Heating period) from 15 Oct to 23 Nov 2016.

| Species | Period I (15-30 Oct) Minor BB | Period II (31 Oct) Intensive BB | | Period III (1 -14, Nov) Major BB | | Period IV (15 -23, Nov) Heating period | |
|---|---|---|---|---|---|---|---|
| | Average concentration | Average concentration | Ratio* | Average concentration | Ratio* | Average concentration | Ratio* |
| $PM_{2.5\text{-cal}}$ | $81.0 \pm 44.5$ | 235 | 2.91 | $163 \pm 46.7$ | 2.01 | $189 \pm 83.0$ | 2.33 |
| Levoglucosan | $0.36 \pm 0.14$ | 4.37 | 12.1 | $0.90 \pm 0.37$ | 2.50 | $0.96 \pm 0.63$ | 2.67 |
| Mannosan | $0.015 \pm 0.005$ | 0.18 | 12.0 | $0.038 \pm 0.015$ | 2.53 | $0.050 \pm 0.026$ | 3.33 |
| OC | $16.2 \pm 7.52$ | 96.3 | 5.93 | $55.2 \pm 17.1$ | 3.41 | $69.4 \pm 24.6$ | 4.28 |
| EC | $12.2 \pm 5.85$ | 36.0 | 2.96 | $25.5 \pm 10.1$ | 2.09 | $36.4 \pm 21.5$ | 2.98 |
| TC | $28.4 \pm 13.1$ | 132 | 4.66 | $80.9 \pm 34.6$ | 2.85 | $106 \pm 55.3$ | 3.73 |
| $SO_4^{2-}$ | $10.3 \pm 8.96$ | 4.56 | 0.44 | $11.8 \pm 6.02$ | 1.15 | $9.08 \pm 3.87$ | 0.88 |
| $NO_3^-$ | $16.6 \pm 12.9$ | 18.1 | 1.09 | $16.5 \pm 6.42$ | 0.99 | $12.6 \pm 5.76$ | 0.76 |
| $NH_4^+$ | $10.1 \pm 7.40$ | 10.0 | 0.99 | $12.0 \pm 4.35$ | 1.19 | $10.3 \pm 3.62$ | 1.02 |
| $K^+$ | $1.16 \pm 0.36$ | 2.61 | 2.25 | $1.76 \pm 0.46$ | 1.52 | $1.65 \pm 0.84$ | 1.42 |
| $Cl^-$ | $3.46 \pm 1.97$ | 7.49 | 2.16 | $5.58 \pm 2.16$ | 1.61 | $6.27 \pm 2.58$ | 1.81 |
| OC/EC | $1.53 \pm 0.35$ | 2.67 | 1.75 | $2.31 \pm 0.59$ | 1.51 | $2.04 \pm 0.31$ | 1.33 |
| $NO_3^-/ SO_4^{2-}$ | $1.74 \pm 0.60$ | 3.96 | 2.28 | $1.50 \pm 0.35$ | 0.86 | $1.42 \pm 0.47$ | 0.82 |
| levoglucosan/OC | $0.025 \pm 0.008$ | 0.045 | 1.80 | $0.016 \pm 0.005$ | 0.64 | $0.014 \pm 0.006$ | 0.56 |
| levoglucosan/EC | $0.039 \pm 0.019$ | 0.121 | 3.10 | $0.038 \pm 0.017$ | 0.97 | $0.028 \pm 0.013$ | 0.72 |
| levoglucosan/ mannosan | $24.9 \pm 4.44$ | 24.1 | 0.97 | $24.8 \pm 6.46$ | 1.00 | $18.3 \pm 4.27$ | 0.73 |
| levoglucosan/$K^+$ | $0.36 \pm 0.081$ | 1.67 | 4.64 | $0.51 \pm 0.16$ | 1.42 | $0.53 \pm 0.15$ | 1.47 |
| NO (ppb) | $21.7 \pm 12.5$ | 21.7 | 1.00 | $39.6 \pm 15.4$ | 1.82 | $39.3 \pm 23.6$ | 1.81 |
| $NO_2$ (ppb) | $21.8 \pm 4.95$ | 26.5 | 1.22 | $32.7 \pm 7.27$ | 1.50 | $24.6 \pm 10.2$ | 1.13 |
| $NO_X$ (ppb) | $43.6 \pm 16.3$ | 48.2 | 1.11 | $72.4 \pm 17.8$ | 1.66 | $64.0 \pm 33.4$ | 1.47 |
| $SO_2$ (ppb) | $5.83 \pm 2.46$ | 8.04 | 1.38 | $11.1 \pm 4.10$ | 1.90 | $9.75 \pm 3.31$ | 1.67 |
| CO (ppm) | $0.44 \pm 0.33$ | 0.70 | 1.59 | $1.65 \pm 0.53$ | 3.75 | $1.18 \pm 0.83$ | 2.68 |
| $O_3$ (ppb) | $9.79 \pm 4.88$ | 23.2 | 2.37 | $7.51 \pm 3.87$ | 0.77 | $9.59 \pm 7.55$ | 0.98 |
| $NH_3$ (ppb) | $14.3 \pm 6.12$ | 11.1 | 0.78 | $18.6 \pm 8.03$ | 1.30 | $21.2 \pm 14.2$ | 1.48 |

*: indicates that the ratios of the heating period, intense BB period or major biomass burning period
were divided by those from the minor BB period.







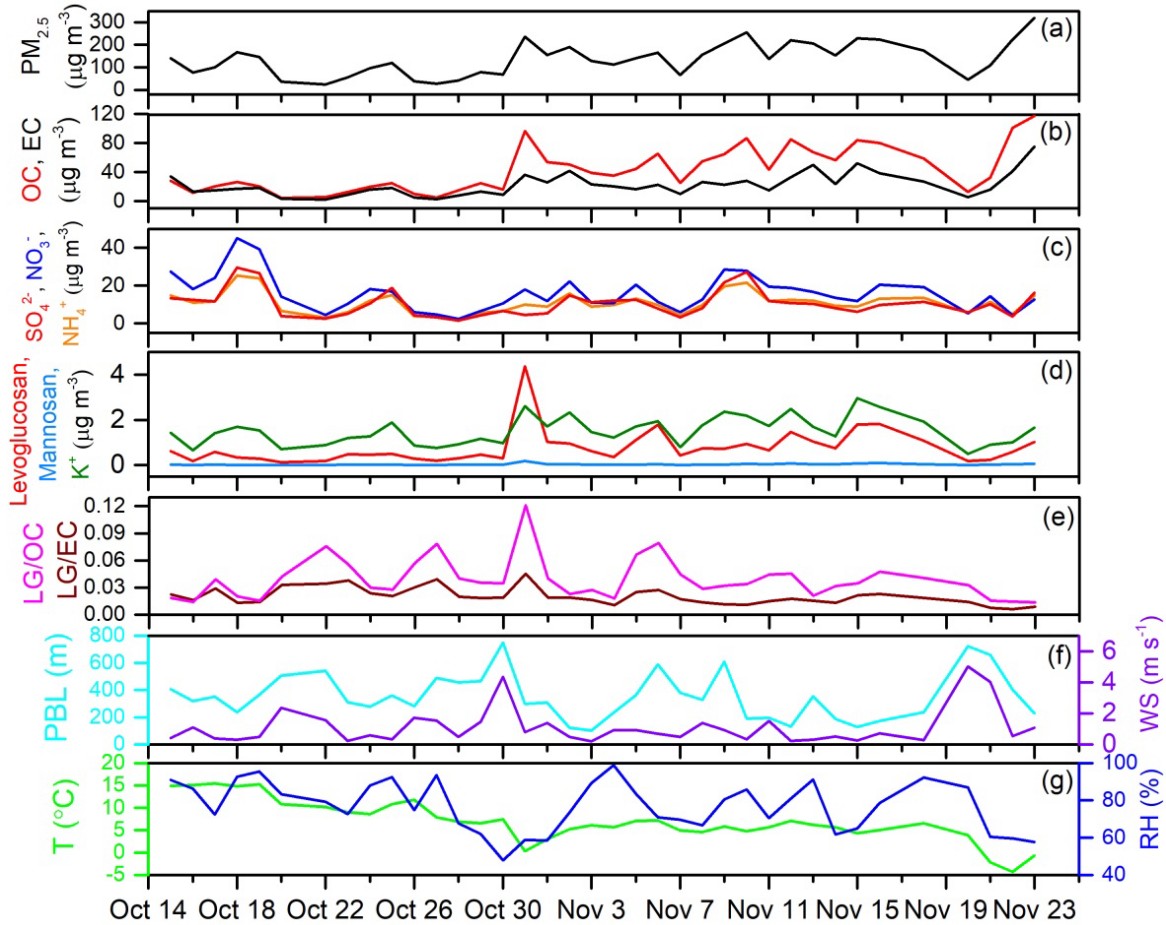

660

**Figure 1.** Time-series variation obtained for $PM_{2.5\text{-cal}}$ and its major components, biomass burning tracers as well as meteorological factors at the GC site during the sampling period from 15 Oct to 23 Nov 2016. (a) $PM_{2.5\text{-cal}}$, (b) OC and EC, (c) secondary inorganic aerosols, i.e., $SO_4^{2-}$, $NO_3^-$ and $NH_4^+$, (d) levoglucosan, mannosan and $K^+$, (e) ratios of levoglucosan to OC (LG/OC) and levoglucosan to EC (LG/EC), (f) PBL and wind speed (WS), (g) temperature (T) and relative humidity (RH).











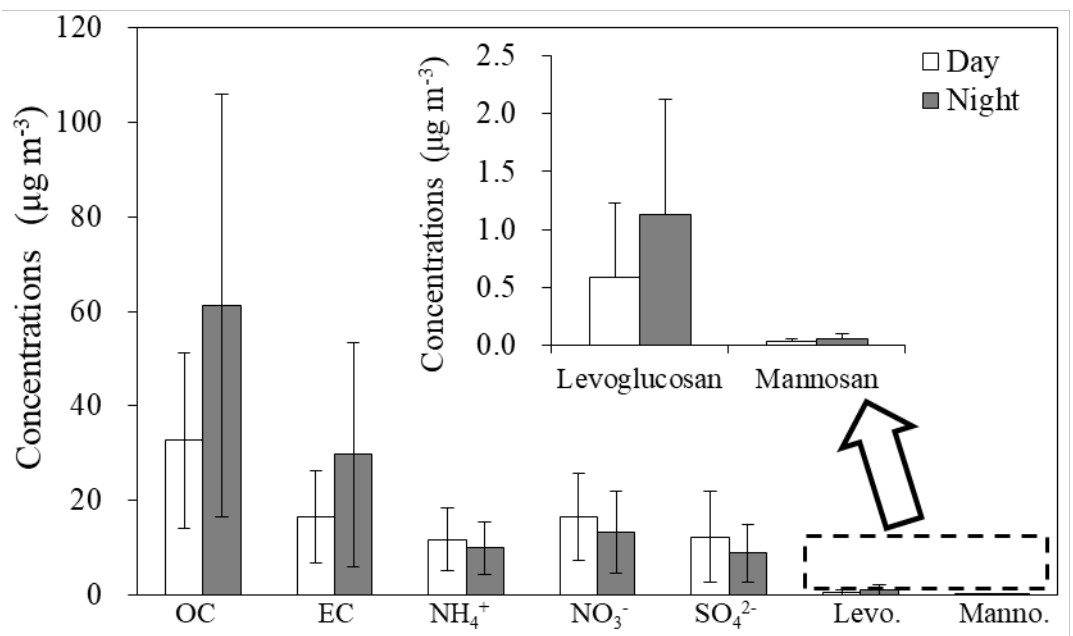


**Figure 2.** Day and night distributions of mean concentrations of main chemical components (OC, EC,
$SO_4^{2-}$, $NO_3^-$ and $NH_4^+$) and biomass burning tracers (levoglucosan and mannosan) in $PM_{2.5}$ observed at
GC site during the sampling period.


















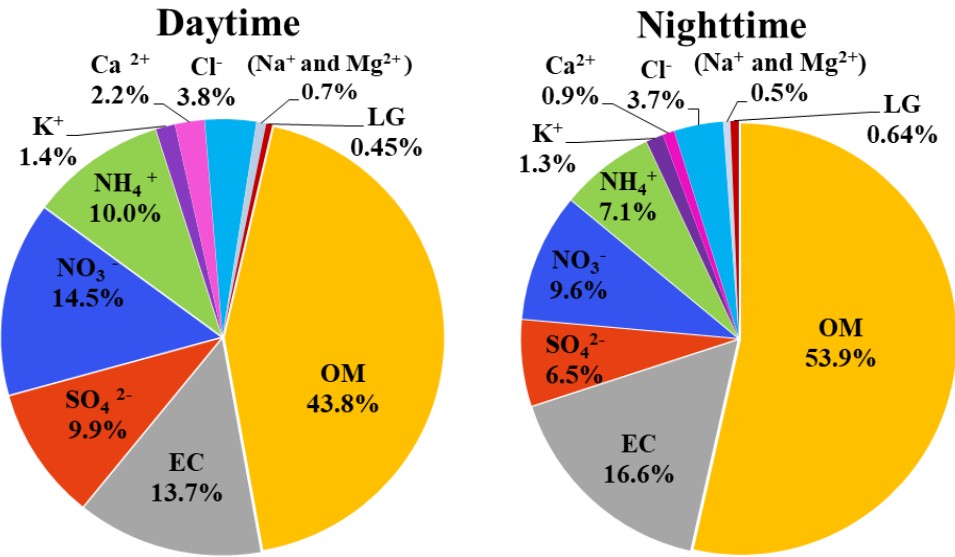

**Figure 3.** Percent contributions of individual component mass concentrations to total estimated $PM_{2.5\text{-}cal}$ mass in daytime and nighttime during the sampling period.


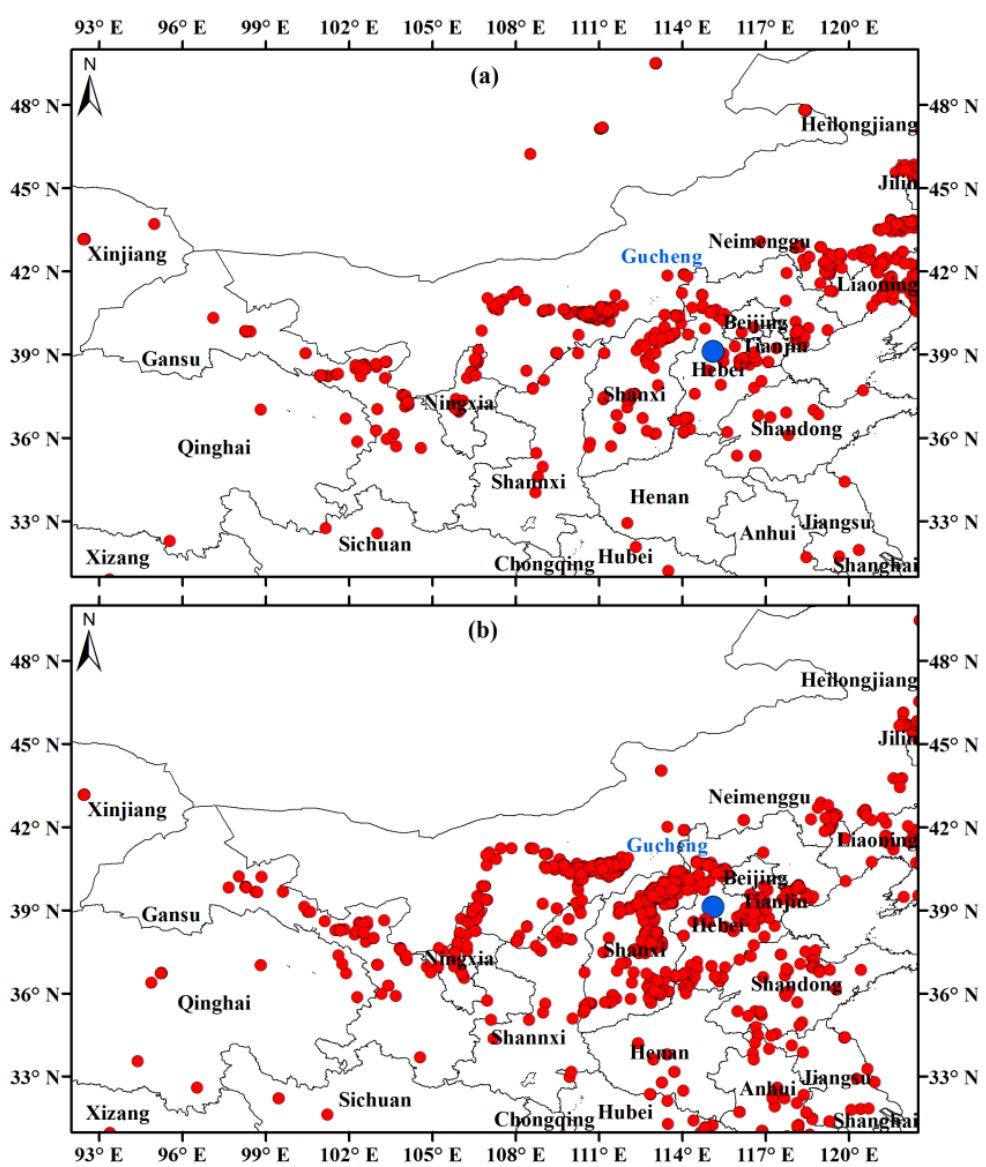

**Figure 4.** Fire spots at GC site and the surrounding provinces from (a) 15-30 October, 2016 and (b) 1 -

23, November, 2016, observed by MODIS Terra satellites (blue dot is GC station).






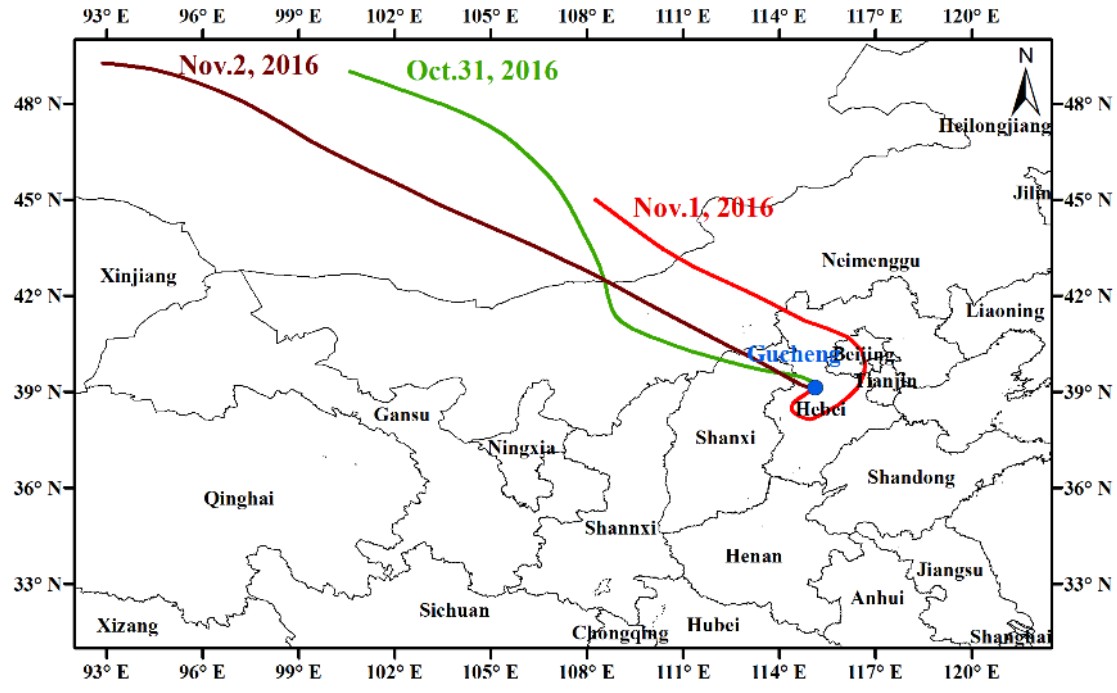

**Figure 5.** 48 h back trajectories at 500 m at GC site (39°09'N, 115°44'E) at 00:00 (UTC time) from 31
October to 2 November, 2016.












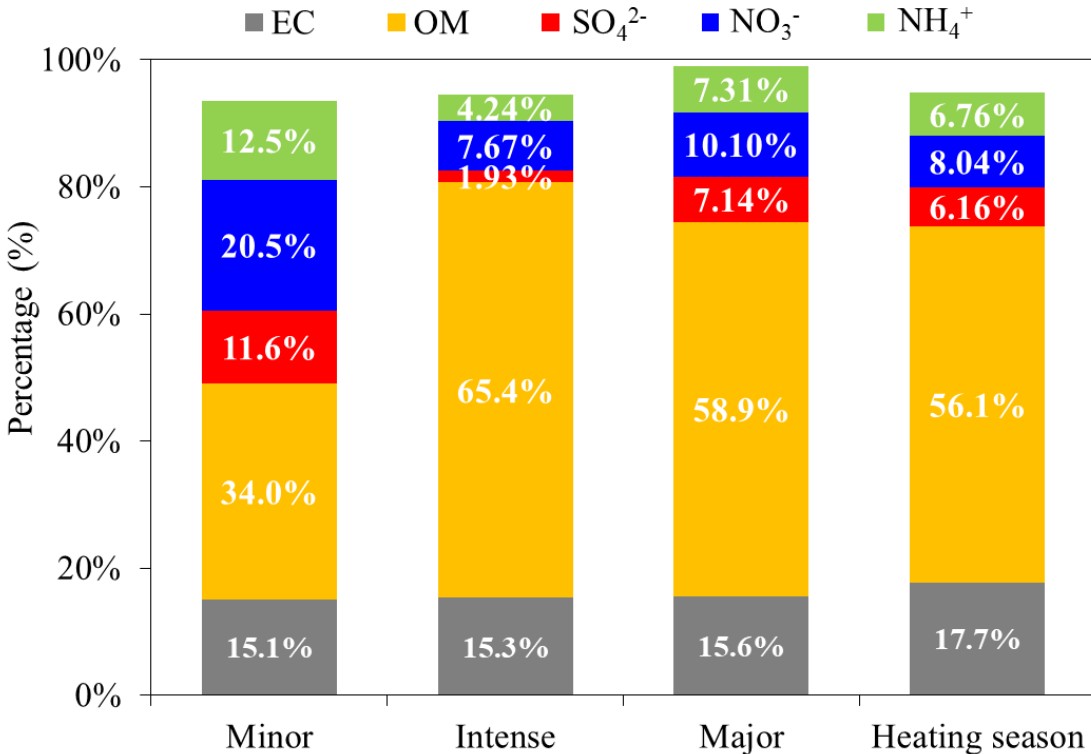


**Figure 6.** Mean percentiles of major components in $PM_{2.5}$ with respect to different biomass burning

pollution periods at GC site during the sampling time.
















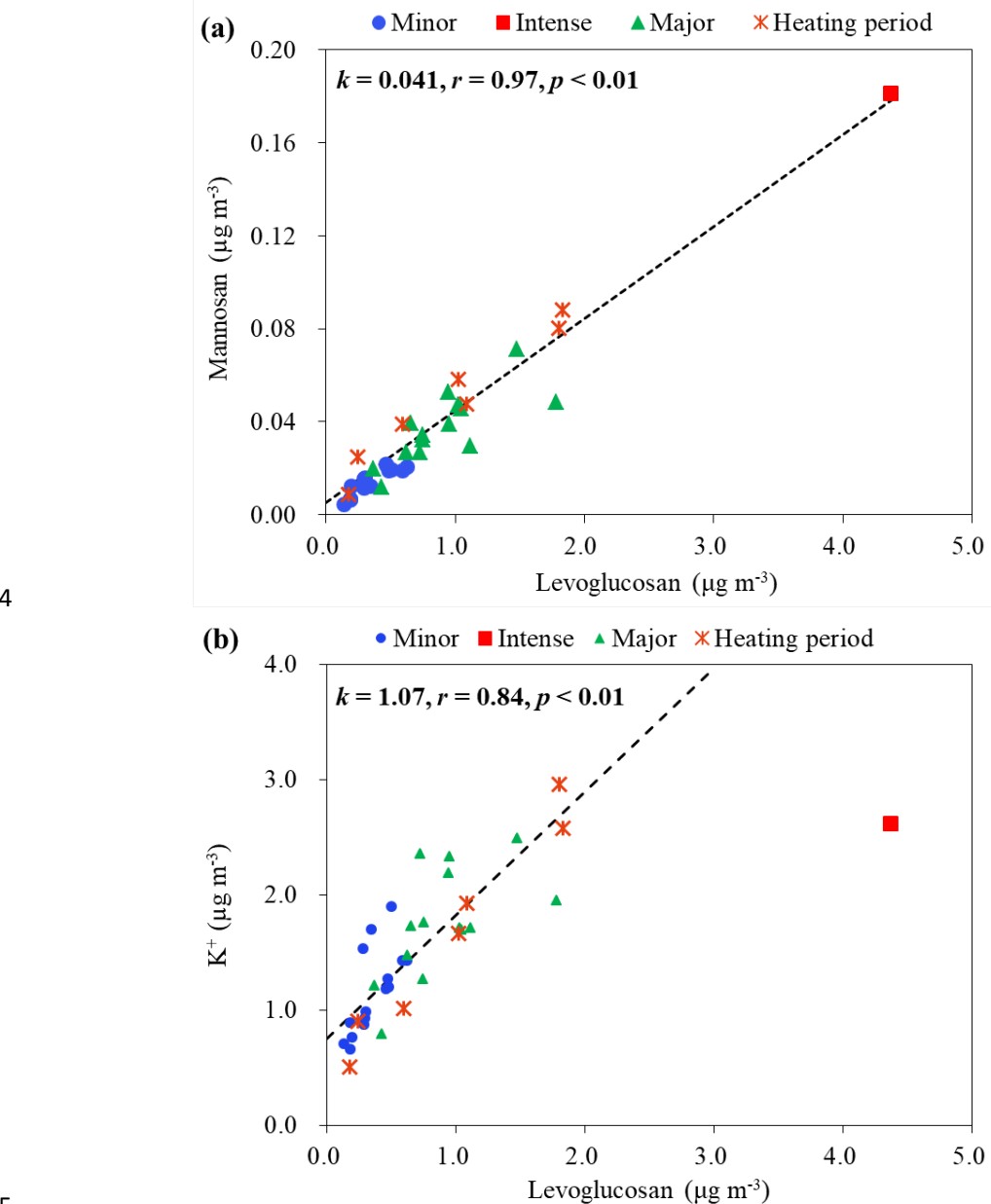



**Figure 7.** Scatter plots of (a) levoglucosan versus mannosan, (b) levoglucosan versus $K^+$. Statistical

analysis of sampling data was conducted with the linear fitting method.





