# Peer review of "Measurement report: Chemical characteristics of PM2.5 during typical biomass"

_Atmospheric Chemistry and Physics, 2020_

## Referee Comment (RC1) · Anonymous Referee #1 · 10 Nov 2020

In this manuscript, the authors report chemical characteristics of PM2.5 under the impact of biomass burning (BB) in the North China Plain. A unique episode with extreme biomass burning impact, with daily concentrations of levoglucosan as high as 4.37 $\mu$g m-3 was captured. The formation process and chemical characteristics of this severe biomass burning pollution episode were also reported. This field measurement was interesting and the data in this study was valuable. This study matches the definition of Measurement Report quite well, presenting substantial new results from field measurements of atmospheric properties and processes. The manuscript is well organized and concisely written, and minor revisions indicated below are needed before publication.

Major comments: (1) LOD (limit of detection) of the water-soluble inorganic ion analysis also suggested described in the experimental section. (2) Experimental section should include more detailed information regarding statistical analysis conducted. (3) "Concentration" in table 1 should be changed to "Average concentration". (4) The meteorological factors (temperature (T), relative humidity (RH), wind speed (WS) and rainfall) in Figure 1 were together expressed in one figure, difficult to distinguish. It is suggested to separate these meteorological factors to two figures and add the time-series variation of PBL as well. (5) The English grammar and usage should be polished by some English native speakers. (6) The abbreviation such as LG and MN is not generally used in literatures. These abbreviations are not easy to be remembered and make the manuscript difficult to understand. I suggest that the authors using the origin names or abbreviations more easily to be remembered. (7) discussion of the possible degradation of levoglucosan should be included in the Day and night distributions. (8) more time series of diagnostic ratio such as levoglucosan to OC ratios should be presented to illustrate the impact of BB

––––––––––––––––––––––––––––––––

---

## Referee Comment (RC2) · Anonymous Referee #2 · 18 Nov 2020

This study reports a measurement research on the characteristics of the chemical components of $PM_{2.5}$ during 15 October to 30 November at the agricultural site of the NCP. The authors linked their results to the BB emission and claimed the importance of softwood burning to the air quality in NCP during the heating season. Overall, this is a nice piece of paper with clear objectives and methods and will provide valuable results. Before considering publication in ACP, major revisions should be made. Some comments and suggestions are listed as follows:

General comment:

Although it is a measurement report, which should present substantial new results from measurements of atmospheric properties and processes, the scientific goal should be improved well through focusing on the innovation in measurement or data analysis methods. The current results are no longer new compared with that reported in 2013 of Beijing by Cheng et al. (2013). What is the current data in this rural site of NCP may bring us to a new knowledge of chemical characteristics, especially in atmospheric properties and processes? Is there any difference between this study with that reported previously, e.g., a faster conversion rate, a new emission type due to the emission control by the government, etc. Besides, the logicality of this paper should be improved. For example, "the LG/MN ratios from crop residue

burning, i.e., rice straw, wheat straw, and other straws, were similar and characterized by high values, yet overlapped with those from hard wood and leaf burning (>10.0), while soft wood characterized by relatively lower LG/MN ratios (< 5.0)". The ratio of LG/MN in this study is around 20, which the authors claim that the air quality was influenced by softwood emission. This conclusion is obviously inconsistent with the their previous analysis.

Specific comments:

1. P4, L107. The abbreviation LG and MN should be spelled out first time. Similar with that in P7, L189, "Elemental carbon and primary organic components", which has been used as EC or POC before. The abbreviation through out the manuscript should be checked carefully to unified.

2. P8, L202. "Moreover, such an enhancement in secondary transformations during daytime is more evident in terms of the mass contributions of secondary inorganic ions to PM2.5-cal, that the contributions of SO42-, NO3- and NH4+ to PM2.5-cal decreased from daytime (9.9%, 14.5% and 10.0%) to nighttime (6.5%, 9.6% and 7.1%) (Figure 3)." The conversion rate of SOR, NOR should be useful here.

3. P8, L214. The BB episodes section. The detailed description of this episode 31 Otc is helpful to readers for understanding, such as the meteorological conditions, wind rose plot. Besides, the PMF or model

simulation should be made to conclude how much the BB contribute to the PM2.5.

4. P9, L230. "The central heating systems in North China cities were operated during period IV, and the ambient level of LG was observed at $0.96 \pm 0.63$ µg m-3, which was slightly higher than that in period III." Is this statement tell us the central heating systems used in NCP will emitted more LG. As we know, the heating system was changed since 2016 over NCP from coal to gas at least in the main cities of NCP. The rest area of NCP are substituted by the electric power system such as air conditioner. Does that means the LG may originated from gas or other fuels?

5. Conclusion section. The local soft wood contributed to high concentrations of PM2.5 in NCP during heating season should be more considered.

6. Language improvement should be made by a native speaker.

---

## Referee Comment (RC3) · Anonymous Referee #3 · 9 Dec 2020

This is a well-written and structured manuscript to discuss the biomass burning pollution status in rural atmosphere of North China by presenting the biomass burning tracers and secondary inorganic ions in PM2.5 during a transition heating season. It is interesting that an episode with extreme biomass burning tracer levels was identified to present the severity of biomass burning pollutions. Biomass burning tracer ratios were also introduced to discuss the biomass source types and burning process. I agree with the data discussion and to publish on ACP. There are some minor errors are necessary to be revised before publishing.

Specific comments:

Line 103: Are the "6 whole-day samples" are used in the data analysis? Please make a note for the "Whole period, N=37" in table 1 to explain sample categories in the data analysis.

Line 153: Why PM2.5measured (measured with High volume sampler) data was not used instead of PM2.5-cal?

Line 163: Organic matter (OM) appears first time in the paper to show the OM contribution to PM2.5-cal. I suggest to explain that how OM was calculated.

Line 170: Please show the data range in these references during summer and winter seasons to give a better understanding how high levels the anhydrosugars were.

Line 199: The contribution of LG to PM2.5-cal during daytime in Figure 3 was 0.45%. Please check the data.

Line 202: Please insert references for the photochemical formation of secondary inorganic species.

Line 234: In Table 2, the OC contribution during intensive BB period II was 96.3, but not 59.9. Please check the data.

Line 276: Please insert the increasing range of OC fraction.

Line 286: Check the data in Figure 6, the SO42- and NO3- contributions during the intense BB episode were 1.93 and 7.67%.

Line 295: The range of LG/MN ratios from crop residue burning in source emission studies is helpful to understand the biomass types.

Line 304: The LG/K+ ratio during III in Table 2 was 0.51, please check the data.

Please also note the supplement to this comment:
https://acp.copernicus.org/preprints/acp-2020-1006/acp-2020-1006-RC3-supplement.pdf

[Figure]

**Supplement:**

**Comments on "Measurement report: Chemical characteristics of PM$_{2.5}$ during typical biomass burning season at an agricultural site of the North China Plain" by Liang et al.**

This is a well-written and structured manuscript to discuss the biomass burning pollution status in rural atmosphere of North China by presenting the biomass burning tracers and secondary inorganic ions in PM$_{2.5}$ during a transition heating season. It is interesting that an episode with extreme biomass burning tracer levels was identified to present the severity of biomass burning pollutions. Biomass burning tracer ratios were also introduced to discuss the biomass source types and burning process. I agree with the data discussion and to publish on ACP. There are some minor errors are necessary to be revised before publishing.

Specific comments:

Line 103: Are the "6 whole-day samples" are used in the data analysis? Please make a note for the "Whole period, N=37" in table 1 to explain sample categories in the data analysis.

Line 153: Why PM$_{2.5measured}$ (measured with High volume sampler) data was not used instead of PM$_{2.5\text{-}cal}$?

Line 163: Organic matter (OM) appears first time in the paper to show the OM contribution to PM$_{2.5\text{-}cal}$. I suggest to explain that how OM was calculated.

Line 170: Please show the data range in these references during summer and winter seasons to give a better understanding how high levels the anhydrosugars were.

Line 199: The contribution of LG to $PM_{2.5-cal}$ during daytime in Figure 3 was 0.45%. Please check the data.

Line 202: Please insert references for the photochemical formation of secondary inorganic species.

Line 234: In Table 2, the OC contribution during intensive BB period II was 96.3, but not 59.9. Please check the data.

Line 276: Please insert the increasing range of OC fraction.

Line 286: Check the data in Figure 6, the $SO_4^{2-}$ and $NO_3^-$ contributions during the intense BB episode were 1.93 and 7.67%.

Line 295: The range of LG/MN ratios from crop residue burning in source emission studies is helpful to understand the biomass types.

Line 304: The LG/$K^+$ ratio during III in Table 2 was 0.51, please check the data.

---

## Author Comment (AC1) · 11 Jan 2021

**Title: "Measurement report: Chemical characteristics of PM2.5 during typical biomass burning season at an agricultural site of the North China Plain"**

**Anonymous Referee #1**

**General Comments:**

In this manuscript, the authors report chemical characteristics of $PM_{2.5}$ under the impact of biomass burning (BB) in the North China Plain. A unique episode with extreme biomass burning impact, with daily concentrations of levoglucosan as high as 4.37 µg m$^{-3}$ was captured. The formation process and chemical characteristics of this severe biomass burning pollution episode were also reported. This field measurement was interesting and the data in this study was valuable. This study matches the definition of Measurement Report quite well, presenting substantial new results from field measurements of atmospheric properties and processes. The manuscript is well organized and concisely written, and minor revisions indicated below are needed before publication.

**Our reply:**  We thank the reviewer for the pertinent comments. We have prepared the point-by-point responses to address the reviewer's comments as shown below. The blue color text shows the amended sections in the manuscript. The line numbers correspond to those in the revised version of the manuscript.

**Major comments:**

(1) **LOD (limit of detection) of the water-soluble inorganic ion analysis also suggested described in the experimental section.**

**Our reply:** According to the referee's comment, LOD (limit of detection) of the water-soluble inorganic ion analysis is described in the experimental section.

"The quartz filter samples were also analyzed for water-soluble inorganic ions by a Dionex

ICS-5000+ ion chromatograph, including $SO_4^{2-}$, $NO_3^-$, $NH_4^+$, $Cl^-$, $Ca^{2+}$, $Na^+$, $K^+$ and $Mg^{2+}$. The method detection limits for the individual ionic species were 0.18 µg $L^{-1}$, 0.15 µg $L^{-1}$, 0.03 µg $L^{-1}$, 0.048 µg $L^{-1}$, 0.08 µg $L^{-1}$, 0.01 µg $L^{-1}$, 0.01 µg $L^{-1}$, 0.008 µg $L^{-1}$, respectively." (See Lines 115-118)

**(2) Experimental section should include more detailed information regarding statistical analysis conducted.**

Our reply: According to the referee's comment, we added the description of statistical methods applied to our data in the revised manuscript.

"Statistical analysis of data, i.e., the correlation analysis between the concentrations of levoglucosan, mannosan and $K^+$ at Gucheng site during the sampling period were conducted with the linear fitting method." (See Lines 151-154)

**(3) "Concentration" in table 1 should be changed to "Average concentration".**

Our reply: According to the referee's comment, we changed "Concentration" to "Average concentration" in Table 1 in the revised paper.

**(4) The meteorological factors (temperature (T), relative humidity (RH), wind speed (WS) and rainfall) in Figure 1 were together expressed in one figure, difficult to distinguish. It is suggested to separate these meteorological factors to two figures and add the time-series variation of PBL as well.**

Our reply: We thank the anonymous referee for this valuable comment. We added the time-series variation of PBL and separated the meteorological factors into two figures, i.e., Figure 1f and Figure 1g.

[Figure]

**Figure 1.** Time-series variation obtained for PM$_{2.5\text{-cal}}$ and its major components, biomass burning tracers as well as meteorological factors at the GC site during the sampling period from 15 Oct to 23 Nov 2016 (a) PM$_{2.5\text{-cal}}$, (b) OC and EC, (c) secondary inorganic aerosols, i.e., SO$_4^{2-}$, NO$_3^-$ and NH$_4^+$, (d) levoglucosan, mannosan and K$^+$, (e) ratios of levoglucosan to OC (LG/OC) and levoglucosan to EC (LG/EC), (f) PBL and wind speed (WS), (g) temperature (T) and relative humidity (RH).

**(5) The English grammar and usage should be polished by some English native speakers.**

**Our reply:** According to the referee's comment, we have improved the English writing in the revised paper.

**(6) The abbreviation such as LG and MN is not generally used in literatures. These abbreviations are not easy to be remembered and make the manuscript difficult to understand. I suggest that the authors using the origin names or abbreviations more easily to be remembered.**

**Our reply:** According to reviewer's suggestion, the abbreviations of LG and MN were changed to

the original names, i.e., levoglucosan and mannosan in the revised manuscript.

**(7) Discussion of the possible degradation of levoglucosan should be included in the Day and night distributions.**

**Our reply:** According to the referee's comment, we added a remark that the chemical degradation of levoglucosan may occur due to photochemical reaction in the ambient aerosols during daytime in the revised paper, extending the discussion of day-night distribution results.

"Moreover, besides the influence from variations of the PBL height, the chemical degradation of levoglucosan may occur due to photochemical reaction in the ambient aerosols during daytime, further enlarging the gap of levoglucosan levels between daytime and nighttime (Sang et al., 2016; Gensch et al., 2018). Consequently, the contribution of levoglucosan to $PM_{2.5-cal}$ during nighttime (0.64%) was observed to be higher than that during daytime (0.37%) (Figure 3)." (See Lines 201-206)

**(8) More time series of diagnostic ratio such as levoglucosan to OC ratios should be presented to illustrate the impact of BB**

**Our reply:** We thank the referee for this valuable comment. We added the time series of levoglucosan to OC ratios as Figure 1e, illustrating the impact of biomass burning. Meanwhile, the discussion of the influence of biomass burning emission on organic aerosol was also updated in the revised paper.

"The levoglucosan/OC ratio was utilized to estimate the effect of biomass burning on ambient organic aerosols. Accordingly, levoglucosan/OC ratios sharply increased to 0.045 during period II, which was noticeably higher than during other periods in this study. Moreover, this level is also higher than most of the published field observations, i.e., at urban sites (Zhang et al., 2008; Cheng et al., 2013; Zhang et al., 2014), rural sites (Sang et al., 2013; Ho et al., 2014; Pietrogrande et al., 2015; Mkoma et al., 2013) and agricultural sites (Ho et al., 2014; Jung et al., 2014), yet lower than at an urban site in northern Italy during wintertime (in the range of 0.01 to 0.13) (Pietrogrande et al., 2015). This illustrates that biomass combustion played an important role in organic aerosol pollution during the intensive BB episode II. However, due to other emissions of OC enhanced during the major BB episode (period III)

and heating season (period IV), i.e., combustion of coal and biofuel for heating, OC increased to an even higher level ($55.2 \pm 17.1$ µgC m$^{-3}$ and $69.4 \pm 24.6$ µgC m$^{-3}$, respectively). Due to the abundance of organic aerosols, the contribution from biomass burning emission was thereby reduced and the levoglucosan/OC ratios during periods III and IV decreased to $0.016 \pm 0.005$ and $0.014 \pm 0.006$, respectively, even lower than those observed in the minor BB period I ($0.025 \pm 0.008$)." (See Lines 254-268)

---

## Author Comment (AC2) · 11 Jan 2021

**Title: "Measurement report: Chemical characteristics of PM$_{2.5}$ during typical biomass burning season at an agricultural site of the North China Plain"**

**Anonymous Referee #2**

This study reports a measurement research on the characteristics of the chemical components of PM$_{2.5}$ during 15 October to 30 November at the agricultural site of the NCP. The authors linked their results to the BB emission and claimed the importance of softwood burning to the air quality in NCP during the heating season. Overall, this is a nice piece of paper with clear objectives and methods and will provide valuable results. Before considering publication in ACP, major revisions should be made. Some comments and suggestions are listed as follows:

**Our reply:** We appreciate the valuable comments of the anonymous referee. We have prepared the point-by-point responses to address the reviewer's comments as shown below. The blue color text shows the amended sections in the manuscript. The line numbers correspond to those in the revised version of the manuscript.

**General comment:**

**(1) Although it is a measurement report, which should present substantial new results from measurements of atmospheric properties and processes, the scientific goal should be improved well through focusing on the innovation in measurement or data analysis methods. The current results are no longer new compared with that reported in 2013 of Beijing by Cheng et al. (2013). What is the current data in this rural site of NCP may bring us to a new knowledge of chemical characteristics, especially in atmospheric properties and processes? Is there any difference between this study with that reported previously, e.g., a faster conversion rate, a new emission type due to the emission control by the government, etc.**

**Our reply:** In fact, the topic of our paper is different from Cheng et al. (2013). Cheng et al. (2013)

focused on investigating the relationships between levoglucosan and other biomass burning tracers (i.e., water soluble potassium and mannosan) based on both ambient samples collected in Beijing and pure biomass burning source samples. And they concluded that there are representative ranges of the levoglucosan to $K^+$ and levoglucosan to mannosan ratios for different kinds of biomass, and they compared the results from the ambient samples collected in Beijing. In section 3.4 we apply their results to our study, i.e., representative ranges of the levoglucosan to $K^+$ and the levoglucosan to mannosan ratios for different kinds of biomass, to discuss the sources for the severe biomass burning event at the rural site in North China. Although, the phenomenon observed in our study on biomass sources identification (section 3.4) is partly similar to those ambient observation results from Beijing during wintertime (Cheng et al., 2013), the discussion of potential influence factors on the biomass burning tracer ratios is different and extended to, e.g., combustion conditions (smoldering and flaming burns), back trajectory analysis, fire activity data and synoptic condition discussion, which were included in our study but not mentioned in Cheng et al. (2013). Moreover, our manuscript also includes the discussion on day-night variations in the patterns of $PM_{2.5}$ chemical components as well as the influence of different levels of biomass combustion pollution on $PM_{2.5}$ chemical characteristics.

Overall, the most notable merits of our manuscript include:

① To the best of our knowledge, this study is the first one to characterize the biomass burning pollution status at a heavily polluted rural site in Hebei province during the autumn-winter transition season, following the corn harvest. The results can provide valuable information about the biomass burning activities in the entire North China region. Moreover, we captured a unique episode with extreme biomass burning pollution, with concentrations of levoglucosan as high as 4.37 μg m$^{-3}$. Based on the multi-analysis of biomass burning molecular tracers, back trajectory analysis, fire activity data and synoptic condition, the formation process and chemical character of this severe biomass burning pollution episode were discussed in detail.

② Combined with other chemical components analysis, our study revealed the different levels of biomass combustion pollution impacting the different types of chemical components in ambient aerosol, which have rarely been reported by previous work.

③ From our observations and those reported in literature, we highlight that both biomass types and combustion conditions (flaming versus smoldering) exert non-negligible impact on the formation mechanisms of biomass burning tracers in the ambient aerosols.

**(2) Besides, the logicality of this paper should be improved. For example, "the LG/MN ratios from crop residue burning, i.e., rice straw, wheat straw, and other straws, were similar and characterized by high values, yet overlapped with those from hard wood and leaf burning (>10.0), while soft wood characterized by relatively lower LG/MN ratios (< 5.0)". The ratio of LG/MN in this study is around 20, which the authors claim that the air quality was influenced by softwood emission. This conclusion is obviously inconsistent with their previous analysis.**

Our reply: Indeed, the levoglucosan/mannosan ratios from hard wood, leaf as well as pure crop residues burning, i.e., rice straw, wheat straw, and other straws, were characterized by high values (>10.0), while pure soft wood is characterized by relatively lower levoglucosan/mannosan ratios (<5.0). The levoglucosan/mannosan ratios during minor, intense, major biomass burning pollution and heating season periods in this study were observed at high values, i.e., 24.9, 24.1, 24.8 and 18.3, respectively. However, compared to the levoglucosan/mannosan ratios during the former three episodes (24.1-24.9, averaged at 24.6), the ratio observed during the heating season period (18.3) decreased by 25.6%. We speculate this decline trend of levoglucosan/mannosan ratios was partly influenced by the raised proportion of softwood combustion for heating, which is characterized by relatively lower levoglucosan/mannosan ratios. In fact, biomass, especially of crop residues (e.g., wheat and corn straw) is more commonly used as biofuel for cooking in the rural areas in North China. However, due to the burning of crop residues or leaves typically being subject to quick flaming combustion under high temperature burning condition, such fuels are not suitable for extended heating during the cold season. According to the local habits, softwoods are also commonly used as biofuels for stove heating in North China during wintertime, especially during periods when the use of coal is restricted in the NCP.

Nonetheless, in order to make the description more clearly and also addressing the comments from third reviewer, the discussion on the influence of different types of biomass on the tracer

ratios has been modified in the revised manuscript as shown below:

"Levoglucosan and mannosan showed a good relationship during the entire sampling period (Figure 7a, $r = 0.97$, $p < 0.01$). The levoglucosan/mannosan ratios during minor, intense, major biomass pollution and heating season periods were observed at high values, i.e., 24.9, 24.1, 24.8 and 18.3 respectively (Table 2, Figure 7). Compared to the former three episodes (24.1 to 24.9, averaged at 24.6), the levoglucosan/mannosan ratios during heating season period (18.3) decreased by 25.6%. Based on source emission studies, the levoglucosan/mannosan ratios from crop residue burning, i.e., rice straw, wheat straw, and corn straw, are similar and are characterized by high values (averaged at 29, in the range of 12 to 55) (Zhang et al., 2007; Engling et al., 2009; Cheng et al., 2013; Jung et al., 2014), yet overlapping with those from hard wood (averaged at 28, in the range of 11 to 146) (Bari et al., 2009; Jung et al., 2014) and grass burning ($18.2 \pm 10.2$) (Sullivan et al., 2008), while softwood is characterized by relatively lower levoglucosan/mannosan ratios (averaged at 4.3, in the range of 2.5 to 4.7) (Engling et al., 2006; Cheng et al., 2013; Jung et al., 2014). Subsequently, this declining trend in the levoglucosan/mannosan ratios during the heating season period was partly caused by the higher proportion of softwood combustion, which is characterized by relatively lower levoglucosan/mannosan ratios. According to the local habits, softwoods, e.g. China fir and pine are also commonly used as biofuels for stove heating in North China, since they allow sustained heating duration." (See Lines 315-332)

**Specific comments:**

**(1) P4, L107. The abbreviation LG and MN should be spelled out first time. Similar with that in P7, L189, "Elemental carbon and primary organic components", which has been used as EC or POC before. The abbreviation through out the manuscript should be checked carefully to unified.**

**Our reply:** According to the referee's comment, we checked the manuscript and confirmed that the acronyms were all defined when mentioned for the first time in the text. Considering the other reviewer's suggestion, the abbreviations of LG and MN were changed to the original names, i.e., levoglucosan and mannosan in the revised manuscript.

**(2) P8, L202.** "**Moreover, such an enhancement in secondary transformations during daytime is more evident in terms of the mass contributions of secondary inorganic ions to PM2.5-cal, that the contributions of SO$_4$$^{2-}$, NO$_3$$^-$ and NH$_4$$^+$ to PM$_{2.5\text{-cal}}$ decreased from daytime (9.9%, 14.5% and 10.0%) to nighttime (6.5%, 9.6% and 7.1%) (Figure 3).**" **The conversion rate of SOR, NOR should be useful here.**

**Our reply:** We thank the referee for this valuable comment. We calculated the conversion rate of SOR and NOR in the revised manuscript, and extended the supplement for the evidence of secondary inorganic aerosol transformations enhanced during daytime.

"The mass contributions of secondary inorganic ions to PM$_{2.5\text{-cal}}$, that is the contributions of SO$_4$$^{2-}$, NO$_3$$^-$ and NH$_4$$^+$ to PM$_{2.5\text{-cal}}$, decreased from daytime (9.9%, 14.5% and 10.0%) to nighttime (6.5%, 9.6% and 7.1%) (Figure 3). Such an enhancement in secondary transformations during daytime is more evident in terms of the sulfur and nitrogen oxidation ratios (SOR and NOR, molar ratio of sulfate or nitrate to the sum of sulfate and SO$_2$ or nitrate and NO$_2$), which have been used previously as indicators of secondary transformations (Sun et al., 2013; Zheng et al., 2015). Both SOR and NOR during daytime were higher than those during nighttime (Figure S3), further confirming the elevated secondary formations of sulfate and nitrate during daytime." (See Lines 209-216)

[Figure]

*Figure S3*. Variation of NOR and SOR during daytime and nighttime, respectively. In the box-whisker plots, the boxes and whiskers indicate the 95th, 75th, 50th (median), 25th and 5th percentiles, respectively. □ indicates the mean value.

**(3) P8, L214. The BB episodes section. The detailed description of this episode 31 Oct is helpful to readers for understanding, such as the meteorological conditions, wind rose plot. Besides, the PMF or model simulation should be made to conclude how much the BB contribute to the PM$_{2.5}$.**

Our reply: According to the referee's comment, the meteorological conditions during intense biomass burning episode on 31 October was described in detail in the revised manuscript.

As for the contributions of biomass burning to carbonaceous aerosol and PM$_{2.5}$, we quantified them by the molecular tracer approach and discussed the results in a companion paper, as it would render this paper too long otherwise. Nonetheless, we thank the referee for this valuable comment and have revised the corresponding text as follows.

"Meanwhile, there was significant change in the meteorological conditions, i.e., the wind direction changed from southwesterly to northerly winds (Figure S4). Northerly winds advected cold and dry air masses, with the lowest hourly temperature observed at -5.3 °C (Figure S5). This notable temperature decline before the commencing of the operation of the central heating systems should have caused intense combustion activities for heating purposes at the rural site. Moreover, the synoptic situation on 31 October, 2016 was under weaker turbulence with low PBL height and small wind speeds (Figure 1f). These worsened meteorological conditions would further enhance aerosol accumulation." (See Lines 229-236)

[Figure]

30 October      31 October      01 November

 Wind-rose diagram of hourly wind direction at the GC site during 30 October, 31 October and 1 November 2016, respectively.

[Figure]

 Hourly temperature from 00:00 on 29th October to 00:00 on 3rd November 2016 at the GC site.

**(4) P9, L230. "The central heating systems in North China cities were operated during period IV, and the ambient level of LG was observed at 0.96 ± 0.63 μg m$^{-3}$, which was slightly higher than that in period III." Is this statement telling us the central heating systems used in NCP will emitted more LG. As we know, the heating system was changed since 2016 over NCP from coal to gas at least in the main cities of NCP. The rest area of NCP are substituted by the electric power system such as air conditioner. Does that means the LG may originated from gas or other fuels?**

**Our reply:** Generally speaking, levoglucosan is a unique molecular tracer for biomass burning, formed during pyrolysis of cellulose, and has been the most common molecular tracers for biomass burning emissions, adopted in numerous laboratory and field studies (Simoneit, 1999;Simoneit, 2002;Engling et al., 2009;Gensch et al., 2018;Chantara et al., 2019;Fortenberry et al., 2018). Thus, there should be no levoglucosan emitted from natural gas combustion. Actually, the ambient level of levoglucosan was likely impacted by various factors, such as emission source characteristics, including biomass categories and combustion

conditions, as well as meteorological conditions, e.g., wind speed and direction, the development of the boundary layer, etc. Therefore, the difference in levoglucosan concentrations between the major biomass burning period and central heating period was impacted by all environmental factors, including source emissions and meteorological conditions. However, in order to make the study focus more on data reporting, we removed the speculations regarding the cause for those similar ambient levoglucosan levels during major biomass burning period and central heating period. Nonetheless, we thank the referee for this valuable comment. To make the description more rigorous, we have modified the corresponding text as follow.

"The central heating systems in North China cities were operated during period IV, and the ambient level of levoglucosan was observed at $0.96 \pm 0.63$ µg m$^{-3}$, which was similar to that observed in period III." (See Lines 249-251)

**(5) Conclusion section. The local soft wood contributed to high concentrations of PM$_{2.5}$ in NCP during heating season should be more considered.**

**Our reply:** According to the referee's suggestion, we modified the description of this conclusion, to make the revised paper focus more on the reported data.

"Compared to the other biomass burning episodes, the levoglucosan/mannosan ratios during the heating season period slightly decreased, while levoglucosan/K$^+$ ratios during the intensive BB period were unusually higher than those in the other three biomass burning periods." (See Lines 365-368)

**(6) Language improvement should be made by a native speaker.**

**Our reply:** According to the referee's comment, we have improved the English language in the revised paper.

**References:**

Cheng, Y., Engling, G., He, K.B., Duan, F.K., Ma, Y.L., Du, Z.Y., Liu, J.M., Zheng, M., and Weber, R.J.: Biomass burning contribution to Beijing aerosol, Atmos. Chem. Phys., 13, 7765-7781, https://doi.org/10.5194/acp-13-7765-2013, 2013.

Chantara, S., Thepnuan, D., Wiriya, W., Prawan, S., and Tsai, Y. I.: Emissions of pollutant gases, fine particulate matters and their significant tracers from biomass burning in an open-system combustion chamber, Chemosphere, 224, 407-416, 10.1016/j.chemosphere.2019.02.153, 2019.

Engling, G., Lee, J. J., Tsai, Y.-W., Lung, S.-C. C., Chou, C. C. K., and Chan, C.-Y.: Size-Resolved Anhydrosugar Composition in Smoke Aerosol from Controlled Field Burning of Rice Straw, Aerosol Science and Technology, 43, 662-672, 10.1080/02786820902825113, 2009.

Fortenberry, C. F., Walker, M. J., Zhang, Y., Mitroo, D., Brune, W. H., and Williams, B. J.: Bulk and molecular-level characterization of laboratory-aged biomass burning organic aerosol from oak leaf and heartwood fuels, Atmospheric Chemistry and Physics, 18, 2199-2224, 10.5194/acp-18-2199-2018, 2018.

Gensch, I., Sang-Arlt, X. F., Laumer, W., Chan, C. Y., Engling, G., Rudolph, J., and Kiendler-Scharr, A.: Using delta(13)C of Levoglucosan As a Chemical Clock, Environ Sci Technol, 52, 11094-11101, 10.1021/acs.est.8b03054, 2018.

Simoneit, B. R. T.: Biomass burning — a review of organic tracers for smoke from incomplete combustion, Applied Geochemistry, 129-162, 2002.

Simoneit, J. J. S., C.G. Nolte, D.R. Oros, V.O. Elias, M.P. Fraser, W.F. Rogge, G.R. Cass□: Levoglucosan, a tracer for cellulose in biomass burning and atmospheric particles, Atmospheric Environment, 173-182, 1999.

---

## Author Comment (AC3) · 11 Jan 2021

**Title: "Measurement report: Chemical characteristics of PM2.5 during typical biomass burning season at an agricultural site of the North China Plain"**

**Anonymous Referee #3**

This is a well-written and structured manuscript to discuss the biomass burning pollution status in rural atmosphere of North China by presenting the biomass burning tracers and secondary inorganic ions in $PM_{2.5}$ during a transition heating season. It is interesting that an episode with extreme biomass burning tracer levels was identified to present the severity of biomass burning pollutions. Biomass burning tracer ratios were also introduced to discuss the biomass source types and burning process. I agree with the data discussion and to publish on ACP. There are some minor errors are necessary to be revised before publishing.

**Our reply:** We thank the reviewer for his/her valuable comments. We have prepared the point-by-point responses to address the reviewer's comments as shown below. The blue color texts indicate the amended sections in the manuscript. The line numbers correspond to those in the revised version of the manuscript.

**Specific comments:**

(1) **Line 103: Are the "6 whole-day samples" are used in the data analysis? Please make a note for the "Whole period, N=37" in table 1 to explain sample categories in the data analysis.**

**Our reply:** According to the referee's comment, we have added a note for "Whole period, N=37" in table 1, explaining sample categories in the data analysis.

"Six whole-day samples were included in the data analysis of the "Whole period". (See Line 649)

(2) **Line 153: Why $PM_{2.5}$ measured (measured with High volume sampler) data was not used instead of $PM_{2.5-cal}$?**

**Our reply:** $PM_{2.5}$ samples were collected using a high-volume sampler (Thermo Scientific, MA, USA; the flow rate was 1.13 $m^3$ $min^{-1}$). Quartz fiber filters (8×10 inch, 2500 QAT-UP; Pall Corporation, NY, USA) taken from the same lot were used for the entire sampling campaign. It is difficult to weigh those big filters with typical laboratory balances; thus, there were no measured $PM_{2.5}$ concentration obtained in this study. Actually, the reconstituted $PM_{2.5-cal}$ mass concentration method has been commonly applied by other filed observations, to demonstrate the variation of ambient $PM_{2.5}$ pollution level (Turpin and Lim, 2001;Kanakidou et al., 2005;Cheng et al., 2015).

**(3) Line 163: Organic matter (OM) appears first time in the paper to show the OM contribution to $PM_{2.5-cal}$. I suggest to explain that how OM was calculated.**

**Our reply:** We thank the referee for this valuable comment. We added the definition of OM in the revised manuscript.

"Organic matter (OM), calculated by multiplying OC values with a coefficient of 1.6, was the most abundant PM component, the daily average value of which was 70.4 ± 49.6 μg $m^{-3}$, …" (See Lines 167-169)

**(4) Line 170: Please show the data range in these references during summer and winter seasons to give a better understanding how high levels the anhydrosugars were.**

**Our reply:** According to the referee's suggestion, we added the data range of levoglucosan during summer and winter season observed in Beijing in the reference.

"The ambient concentrations of levoglucosan in this study were higher than those observed in the city of Beijing during the summer (averaged at 0.23 ± 0.37 μg $m^{-3}$, in the range of 0.06 to 2.30 μg $m^{-3}$) and winter (averaged at 0.59 ± 0.42 μg $m^{-3}$, in the range of 0.06 to 1.94 μg $m^{-3}$) of 2010-2011 (Cheng et al., 2013)." (See Lines 173-176)

**(5) Line 199: The contribution of LG to $PM_{2.5-cal}$ during daytime in Figure 3 was 0.45%. Please check the data.**

**Our reply:** We thank the referee for this valuable comment. We checked the data and confirmed

that the contribution of levoglucosan to $PM_{2.5\text{-cal}}$ during daytime was 0.45% and corrected it in the revised manuscript.

"Consequently, the contribution of levoglucosan to $PM_{2.5\text{-cal}}$ during daytime (0.45%) was observed to be considerably lower than that during nighttime (0.64%) (Figure 3)." (See Lines 204-206)

**(6) Line 202: Please insert references for the photochemical formation of secondary inorganic species.**

**Our reply:** According to the referee's suggestion, we added the related references for the photochemical formation of secondary inorganic species in the revised manuscript.

"Thus, the secondary inorganic species ($SO_4^{2-}$, $NO_3^-$ and $NH_4^+$) were enhanced during daytime due to photochemical formation (Sun et al., 2013; Zheng et al., 2015; Wu et al., 2018)." (See Lines 207-209)

**(7) Line 234: In Table 2, the OC contribution during intensive BB period II was 96.3, but not 59.9. Please check the data.**

**Our reply:** We thank the referee for this valuable comment. We checked the data and confirmed that the OC concentration during the intensive BB period II was 96.3 µg m$^{-3}$, and corrected it in the revised manuscript.

"The concentrations of OC and EC were also observed to be strongly elevated in period II (Table 2), and especially OC levels increased to 96.3 µg m$^{-3}$ during the intensive BB episode II, nearly 6 times of those during the minor BB period (16.2 ± 7.52 µg m$^{-3}$)." (See Lines 252-254)

**(8) Line 276: Please insert the increasing range of OC fraction.**

**Our reply:** According to the referee's comment, we added the increasing range of the OC fraction in the revised paper.

"…, while the OC fraction increased significantly from 34.0% during the minor BB period I to 65.4% during the intense BB period II." (See Lines 299-300)

**(9) Line 286: Check the data in Figure 6, the SO$_4^{2-}$ and NO$_3^-$ contributions during the intense BB episode were 1.93 and 7.67%.**

Our reply: We thank the referee for this comment. We checked the data and confirmed that the contributions of SO$_4^{2-}$ and NO$_3^-$ to PM$_{2.5\text{-cal}}$ during the intense BB episode were 1.93% and 7.67%, respectively, and corrected them in the revised manuscript.

"The contributions of SO$_4^{2-}$, NO$_3^-$ and NH$_4^+$ to PM$_{2.5\text{-cal}}$ during the minor BB episode (11.6%, 20.5% and 12.5%) substantially declined during the intense BB episode (1.93%, 7.67% and 4.24%)." (See Lines 308-310)

**(10) Line 295: The range of LG/MN ratios from crop residue burning in source emission studies is helpful to understand the biomass types.**

Our reply: According to the referee's comment, we added the findings regarding levoglucosan/mannosan ratios from different biomass burning source emission studies in the revised paper.

"Based on source emission studies, the levoglucosan/mannosan ratios from crop residue burning, i.e., rice straw, wheat straw and corn straw, are similar and are characterized by high values (averaged at 29, in the range of 12 to 55) (Zhang et al., 2007; Engling et al., 2009; Cheng et al., 2013; Jung et al., 2014), yet overlapping with those from hardwood (averaged at 28, in the range of 11 to 146) (Bari et al., 2009; Jung et al., 2014) and grass burning (18.2 ± 10.2) (Sullivan et al., 2008), while softwood is characterized by relatively lower levoglucosan/mannosan ratios (averaged at 4.3, in the range of 2.5 to 4.7) (Engling et al., 2006; Cheng et al., 2013; Jung et al., 2014)." (See Lines 320-327)

**(11) Line 304: The LG/K$^+$ ratio during III in Table 2 was 0.51, please check the data.**

Our reply: We thank the referee for this comment. We checked the data and confirmed that the levoglucosan/K$^+$ ratio during episode III was 0.51, and corrected it in the revised manuscript.

"The levoglucosan/K$^+$ ratios during periods III and IV (0.51 and 0.53) were similar to those during a BB episode at an urban site in Beijing during wintertime (levoglucosan/K$^+$ = 0.51) (Cheng et al., 2013)." (See Lines 335-337)

**Reference:**

Cheng, Y., He, K.-b., Du, Z.-y., Zheng, M., Duan, F.-k., and Ma, Y.-l.: Humidity plays an important role in the PM2.5 pollution in Beijing, Environmental Pollution, 197, 68-75, 10.1016/j.envpol.2014.11.028, 2015.

Kanakidou, M., Seinfeld, J., Pandis, S., Barnes, I., Dentener, F., Facchini, M., Dingenen, R. V., Ervens, B., Nenes, A., and Nielsen, C.: Organic aerosol and global climate modelling: a review, Atmospheric Chemistry and Physics, 5, 1053-1123, 2005.

Turpin, B. J., and Lim, H.-J.: Species Contributions to PM2.5 Mass Concentrations: Revisiting Common Assumptions for Estimating Organic Mass, Aerosol Science and Technology, 35, 602-610, 10.1080/02786820119445, 2001.